# Inverse Design for Fluid-Structure Interactions using Graph Network Simulators

**Kelsey R. Allen** *
DeepMind, UK

**Tatiana Lopez Guevara** *
DeepMind, UK

**Kimberly Stachenfeld** *
DeepMind, UK

**Alvaro Sanchez-Gonzalez**
DeepMind, UK

**Peter Battaglia**
DeepMind, UK

**Jessica Hamrick**
DeepMind, UK

**Tobias Pfaff**
DeepMind, UK

`{krallen,zepolitat,stachenfeld,alvarosg,peterbattaglia,jhamrick,tpfaff}@deepmind.com`

## Abstract

Designing physical artifacts that serve a purpose—such as tools and other functional structures—is central to engineering as well as everyday human behavior. Though automating design using machine learning has tremendous promise, existing methods are often limited by the task-dependent distributions they were exposed to during training. Here we showcase a *task-agnostic* approach to inverse design, by combining general-purpose graph network simulators with gradient-based design optimization. This constitutes a simple, fast, and reusable approach that solves high-dimensional problems with complex physical dynamics, including designing surfaces and tools to manipulate fluid flows and optimizing the shape of an airfoil to minimize drag. This framework produces high-quality designs by propagating gradients through trajectories of hundreds of steps, even when using models that were pre-trained for single-step predictions on data substantially different from the design tasks. In our fluid manipulation tasks, the resulting designs outperformed those found by sampling-based optimization techniques. In airfoil design, they matched the quality of those obtained with a specialized solver. Our results suggest that despite some remaining challenges, machine learning-based simulators are maturing to the point where they can support general-purpose design optimization across a variety of fluid-structure interaction domains.

## 1 Introduction

Automatically designing objects to exhibit a desired property—often referred to as *inverse design*—has the potential to transform science and engineering, including aerodynamics [27], material design [14], optics [21], and robotics [35, 75]. Despite its promise, widespread practice of inverse design has been limited by the availability of fast, general-purpose forward models. Traditionally, many methods rely on specialized "classical" solvers, which are handcrafted to simulate a particular physical process. However, these models can sometimes be slow, may not provide gradients, or in certain domains, might even be unknown [23].

To mitigate these issues, machine learning is playing an increasingly important role by substituting all or part of the inverse design pipeline. "End-to-end" methods can bypass expensive simulation and data collection by learning a mapping directly from design parameters (such as the shape of an airplane wing) to a task objective (such as the observed drag) [59, 76, 19]. Learned generative

---

*These authors contributed equally and are listed in alphabetical order.

36th Conference on Neural Information Processing Systems (NeurIPS 2022).

models or policies can propose candidate design parameters for a given task [52, 17, 19, 24, 12, 47]. These types of methods can be used to solve narrowly defined tasks more efficiently than traditional methods, but come with significant limitations: they require access to a dataset of example designs and tasks for training, and often cannot generalize to tasks or designs outside this distribution [71, 77].

In this work, we focus on a *task-agnostic* approach to inverse design, by employing learned simulator models pre-trained on dynamics with *no* access to the task objective or design space during training, and that, once trained, can be used to solve many new design tasks. This approach, however, requires models with sufficient long-term stability and strong generalization, which commonly applied black-box model architectures such as CNNs often lack [45, 69, 72, 4].

A class of learned physics simulators based on graph neural networks (GNNs) [57, 63] could be a promising candidate for supporting inverse design, as they have shown success in general-purpose physical prediction, exhibiting high accuracy and generalization for a broad range of physics tasks. However, these models have so far mostly been tested for forward prediction alone. To be useful in a design setting, they must also be smooth and robust, as optimizers will exploit unreliable regions in the state space [53]. Moreover, to use scalable gradient-based optimization techniques, simulation models must additionally exhibit gradient stability over extremely large state and action spaces, and over hundreds of time-steps. It is unclear if graph network simulators exhibit these properties.

In this paper, we investigate whether learned GNN simulators can support effective inverse design. We pretrain a GNN simulator on a task-independent dynamics dataset, and at test time perform gradient-based optimization for zero-shot design across a variety of tasks. We perform a large set of experiments and find that (1) the learned simulator model is accurate and flexible enough to support a wide variety of design tasks; (2) it generalizes well enough to support design tasks far outside the training regime; (3) gradients from the learned model turn out to be very useful for inverse design, with gradient-based optimization strongly outperforming a variety of sampling-based optimizers; (4) and surprisingly, gradients remain stable even when rolling the model out for hundreds of steps. In experiments on high-dimensional fluid manipulation (2D FLUID TOOLS and 3D WATERCOURSE) we demonstrate that gradient-based optimization on learned models can find high-quality designs over hundreds of time steps, through states with thousands of particles, in tasks with up to 625 design parameters. The same model matches performance of a specialized solver on aerodynamics design (AIRFOIL) with a much simpler setup, highlighting the flexibility and precision of this approach.

## 2    Related Work

Modern design techniques rely on physical simulators so that designs can be evaluated virtually before being tested physically. **Solving inverse design problems with simulators** has been studied in a variety of disciplines, including nanophotonics [54], material science [25], mechanical design [22], and aerodynamics [2, 60]. Many of these rely on black-box optimization, including active and adaptive sampling methods [68, 3]. In recent years, differentiable simulators [29, 44, 64] have garnered attention, as they allow for more sample-efficient gradient-based optimization. However, like classical solvers, they are typically narrow in application scope, as many simulation techniques are hard to express as a differentiable program (e.g. constraint dynamics and multiphysics coupling). Our work aims to lessen this burden by learning these simulators from data.

**Machine learning for inverse design** replaces or augments the simulation model and/or the optimization procedure with learned modules [e.g. 16, 19, 34, 28, 37, 42, 48, 50, 52, 59, 66, 78]. Generative approaches enable efficient optimization by learning to sample more likely designs (thus calling an expensive simulator fewer times). End-to-end predictive approaches learn a surrogate model that maps designs to task rewards directly (replacing the simulator entirely). This permits not only sampling-based optimization, but also gradient-based techniques when the surrogate model is differentiable [59]. However, surrogate models often generalize poorly to designs that lie outside of their training dataset, so many ML approaches for design must keep the optimization procedure close to the training dataset [12, 30], or introduce alternative objectives for the surrogate model to increase robustness [71, 77]. In all cases, these methods require access to datasets with (design, task reward) pairs on the specific task, making zero-shot generalization to new tasks impossible. Instead, we propose to learn the underlying physical dynamics *independent of the design task*, allowing both the design space and task to be specified at design time.

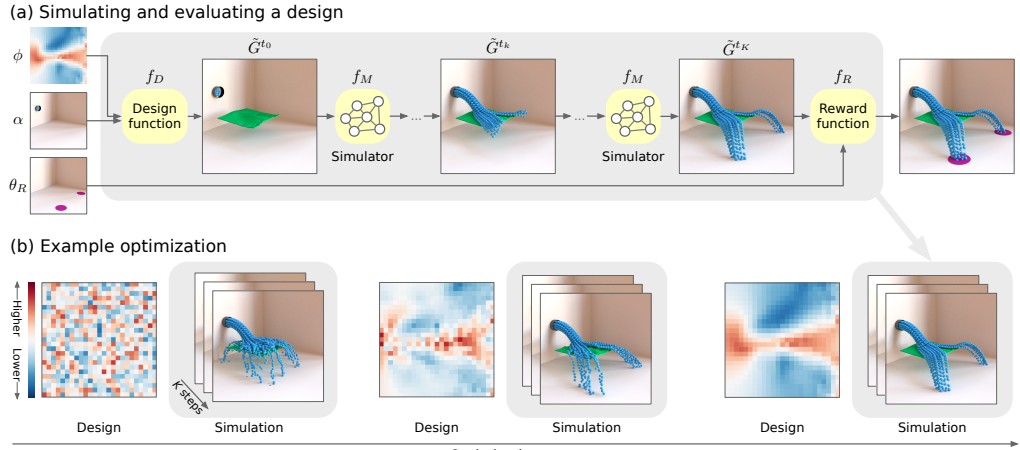

Figure 1: Optimizing a physical design. Here, the goal is to direct a stream of water (shown in blue) into two "pools" (shown in purple) by designing a "landscape" (shown in green) parameterized as a 2D height field. (a) The simulation pipeline takes in a design $\phi$ and initial conditions $\alpha$ and uses the design function $f_D$ to produce an initial state. The simulation is rolled out with a pre-trained learned simulator $f_M$ for $K$ steps, at which point the final state is passed (along with reward parameters $\theta_R$) to the reward function $f_R$, which computes the quality of the design. (b) Each step of optimization involves rolling out the simulation and then adjusting the design ($\phi$) accordingly using an optimizer such as gradient descent or CEM. Shown are selected frames from gradient-based optimization in the *2 Pools* task of the 3D WATERCOURSE domain.

In this way, our approach is related to methods in **model-based reinforcement learning** (MBRL), which learns action-conditioned transition models to support flexible downstream planning and control. However, learned simulators are rarely accurate, stable, and generalizable enough to support better downstream task performance [36, 53]. Techniques from MBRL need to mitigate model error either by learning a global, task-conditioned *policy* during model learning [45] (which then cannot be adapted to novel tasks on-the-fly), or by *re-planning* at every timestep to avoid error accumulation [15]. Because of such sensitivities, these methods have only been demonstrated in relatively small-scale problems, for example with (up to 21 degrees of freedom DOF [10, 62, 73]) or more complex domains with small action spaces (6 DOF [51]), unlike the thousands of state dimensions and hundreds of action dimensions and timesteps we consider here.

We focus on **graph neural networks** as our simulator model, as they have recently demonstrated high accuracy, stability, domain-generality, and generalization performance in the forwards direction for physics domains [7, 57, 63]. However, it is not known whether these simulators can now support gradient-based, high-dimensional inverse design which demands high accuracy, well-behaved gradients, long-term rollout stability, and generalization beyond the training data. Our experimental investigations show that they can, and that particular architectural choices are critical to this success.

## 3   Problem Formulation

Consider the design task depicted in Figure 1, in which the goal is to direct a stream of water (shown in blue) into two "pools" (shown in purple) by designing a "landscape" (shown in green) parameterized as a 2D height field. Here, an ideal design will create ridges and valleys that direct fluid into the two targets. In the next sections, we formalize what it means to find and evaluate such a design and discuss our choices for simulator and optimizer.

**Learned simulators**    To demonstrate the utility of learned simulators for finding physical designs, we rely on the recently developed MESHGRAPHNETS model [57], which is an extension of the Graph Network Simulator (GNS) model for particle simulation [63]. MESHGRAPHNETS is a type of message-passing graph neural network (GNN) that performs both edge and node updates [6, 32], and which was designed specifically for physics simulation by enforcing locality and translation equivariance. We briefly summarize the learned simulator here, and refer interested readers to the original papers for details.

We consider simulations over physical states represented as graphs $G \in \mathcal{G}$. The state $G = (V, E)$ has nodes $V$ connected by edges $E$, where each node $v \in V$ is associated with a position $\mathbf{u}_v$ and additional dynamical quantities $\mathbf{q}_v$. These graphs may be either meshes (as in MESHGRAPHNETS) or particle systems (as in GNS). In a mesh-based system (such as AIRFOIL), $V$ and $E$ correspond to vertices and edges in the mesh, respectively. In a particle system (such as 2D FLUID TOOLS), each node corresponds to a particle and edges are computed dynamically based on proximity. Under this framework, we can also consider hybrid mesh-particle systems (such as 3D WATERCOURSE). The model is constrained to perform local computation, by performing message passing over these edges, and to perform equivariant computations by encoding position as the relative displacement and distance between neighboring particles. Intuitively, this allows it to generalize well to scenarios where the local interactions are the same but the global arrangement of particles changes. See Appendix B for model implementation details, and Appendix C for domain-specific details.

The simulation dynamics are given by a "ground-truth" simulator $f_S : \mathcal{G} \to \mathcal{G}$ which maps the state at time $t$ to that at time $t + \Delta t$. The simulator $f_S$ can be applied iteratively over $K$ time steps to yield a trajectory of states, or a "rollout," which we denote $(G^{t_0}, ..., G^{t_K})$. Using MESHGRAPHNETS, we learn an approximation $f_M$ of the ground-truth simulator $f_S$. The learned simulator $f_M$ can be similarly applied to produce rollouts $(\tilde{G}^{t_0}, \tilde{G}^{t_1}, ..., \tilde{G}^{t_K})$, where $\tilde{G}^{t_0} = G^{t_0}$ represents initial conditions given as input. We note that a learned simulator allows us to take larger time-steps than $f_S$ permitting shorter rollout lengths: in our running example, one model step corresponds to 200 steps of the classical simulator. See Figure 1a for an illustration of simulation using a learned model. To improve stability for longer rollouts, random Gaussian noise was added to the inputs during training.

To optimize a physical design, we leverage the pipeline shown in Figure 1: (1) transform design parameters into an initial scene, (2) simulate the scene using $f_M$ or $f_S$, (3) evaluate how well the simulation achieves the desired behavior, and (4) adjust the design parameters accordingly.

**Design parameters** To produce the initial state $G^{t_0}$, we introduce a differentiable design function $f_D : \Phi \times \mathcal{A} \to \mathcal{G}$ which maps design parameters $\phi \in \Phi$ and other initial conditions $\alpha \in \mathcal{A}$ to an initial state: $G^{t_0} = f_D(\phi, \alpha)$. In Figure 1, $\phi$ is the 2D height field of the mesh, while $\alpha$ is the non-controllable objects in the scene like the initial position of the fluid.

**Maximizing reward** The reward function $f_R : \mathcal{G} \times \Theta_R \to \mathbb{R}$ maps the final state of a length-$K$ trajectory ($G^{t_K}$ or $\tilde{G}^{t_K}$) and reward function parameters $\theta_R \in \Theta_R$ to a scalar value. In our running example, the reward function is defined as the Gaussian likelihood of each fluid particle under the closest "pool", averaged across particles (Figure 1a). Reward function parameters $\theta_R$ are the means and standard deviations of these pools.

We define the objective under the ground-truth simulator $f_S$ as $J_S(\phi) := f_R(f_S^{(K)}(f_D(\phi, \alpha)); \theta_R)$, where $f_S^{(K)}$ indicates $K$ applications of the simulator. We want to find the design parameters that maximize $J_S$, i.e. $\phi^* = \operatorname{argmax}_\phi J_S(\phi)$. We can approximate this optimization using a *learned* simulation model by maximizing $J_M(\phi) := f_R(f_M^{(K)}(f_D(\phi, \alpha)); \theta_R)$. Crucially, multiple reward functions can be optimized with the same model $f_M$.

**Optimizers** Optimal design parameters $\phi^*$ can be found using any generic optimization technique. Given the differentiability of the learned simulator $f_M$, we are particularly interested in evaluating gradient-based optimization, which requires fewer function evaluations and scales better to large design spaces than sampling-based techniques [10]. We use the Adam optimizer [46] to find $\phi^*$ by computing the gradient $\nabla_\phi J_M(\phi)$. This involves backpropagating gradients through the reward function $f_R$, length-$K$ rollout produced by $f_M^{(K)}$, and design function $f_D$.

As a baseline, we consider the cross-entropy method (CEM) [61], a gradient-free sampling-based technique that is popular in model-based control [20, 73]. CEM can be used with any simulator and works by sampling a population of candidates for $\phi$ and evolving them to maximize the reward. However, CEM requires multiple evaluations of $f_M$ or $f_S$ per optimizer iteration (depending on the population size), whereas Adam requires only a single evaluation per iteration. We additionally consider covariance matrix adaptation (CMA-ES) [38] and Bayesian Optimization (BO) [68] in Section D.2 as two additional gradient-free optimizers.

Across our design tasks (Section 4) we compared: gradient descent with Adam on the learned simulator model (**GD-M**), CEM with the learned simulator model (**CEM-M**), and CEM with the

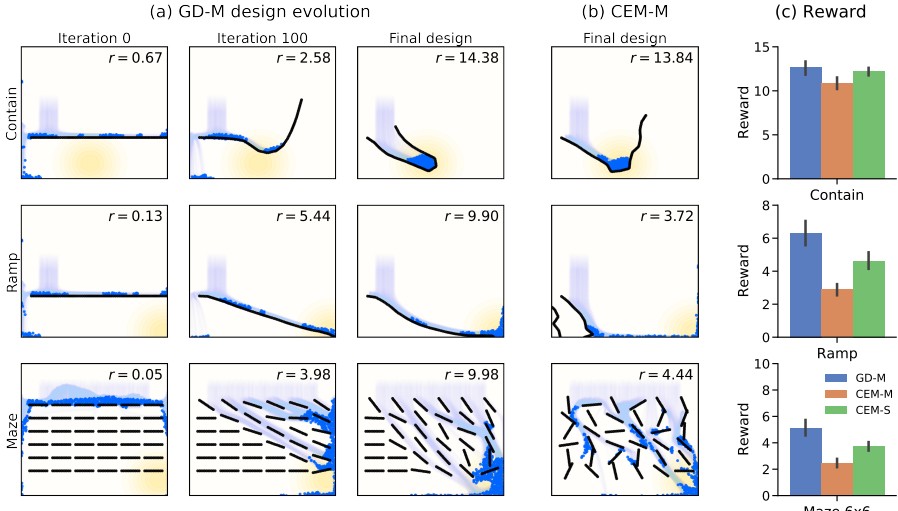

Figure 2: 2D FLUID TOOLS results. The state spaces consist of $10^2$–$10^3$ particles and the design spaces of 16–36 parameters. (a) Evolution of designs found by GD-M during optimization for each 2D FLUID TOOLS task. Visualizations correspond to simulations of the designs under $f_S$. The design is shown in black, fluid particles in blue, and Gaussian reward in yellow. The transparent particles show the location of fluid for $t < t_K$, and the solid particles show the location of fluid at the final frame ($K = 150$). $r$ denotes reward for the current design. (b) Final designs found by CEM-M. (c) Mean reward over 50 reward locations (with bootstrapped $95\%$ confidence intervals) obtained by each optimizer across the 2D FLUID TOOLS tasks. For *Contain* and *Ramp*, results are shown for 16 joints; for *Maze*, results are shown for a $6 \times 6$ grid of 36 rotors. Across these tasks, GD-M outperforms both CEM-M and CEM-S.

ground-truth simulator (**CEM-S**). We compare to further optimizer baselines, such as Bayesian Optimization [68] and CMA-ES [38] in the Appendix (Section D.2). In all tasks, $f_S$ is non-differentiable, preventing a comparison to GD-S. However, in the special case of AIRFOIL, we compare to **DAFoam** [40], a specialized solver which computes gradients with the adjoint method.

**Evaluation**   Unless otherwise noted, we always evaluate the quality of an optimized design $\phi^*$ using the ground-truth objective $J_S(\phi^*)$, regardless of whether $\phi^*$ was found using the learned model $f_M$ (as in CEM-M and GD-M) or the ground-truth simulator $f_S$ (as in CEM-S). We also use rollouts from $f_S$ to produce visualizations in the figures.

## 4   Design Tasks

We formulate a set of design tasks across three different physical domains with high-dimensional state spaces and complex dynamics. Each domain uses a different ground-truth simulator $f_S$, which is used to evaluate designs and pre-train the learned simulator model $f_M$ in a task-agnostic way. During optimization, all the design tasks on each domain share the same pre-trained model.

**2D FLUID TOOLS**   Inspired by existing 2D physical reasoning benchmarks [1, 5], these tasks involve creating one or more 2D "tool" shapes to direct fluid into a particular goal region (Figure 2, Section C.1). We consider three tasks with 4–48 design parameters where the goal is to guide the fluid such that each particle comes as close as possible to the center of a randomly-sampled yellow reward region. In *Contain*, the joint angles $\phi_{\mathrm{joints}}$ of a multi-segment tool must be optimized to catch the fluid by creating cup- or spoon-like shapes. In *Ramp*, the joint angles $\phi_{\mathrm{joints}}$ of a multi-segment tool must be optimized to guide the fluid to a distant location. In *Maze*, the rotation $\phi_{\mathrm{rot}}$ of multiple tools must be optimized to funnel the fluid to the target location. Fluid dynamics are represented using $10^2$–$10^3$ particles, and unrolled for up to 300 time steps.

The learned model $f_M$ is trained on the 2D WaterRamps dataset released in [63] which uses the solver in [43] to generate trajectories. The dataset contains scenes with 1–4 straight line segments; no

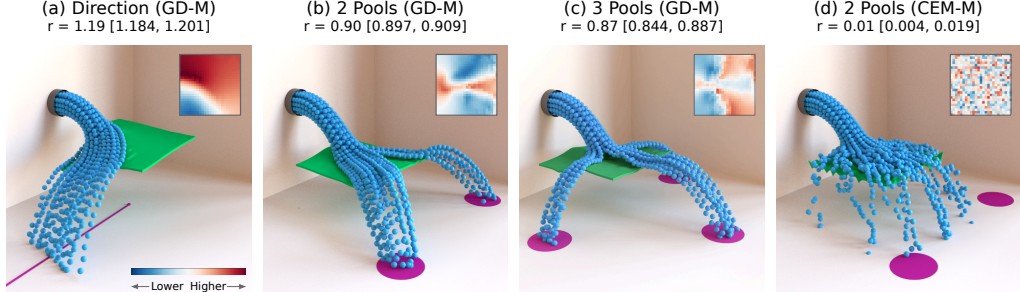

Figure 3: 3D WATERCOURSE results found with GD-M or CEM-M and evaluated with $f_S$. Simulations use up to 2000 particles and 625 design parameters. The heightmap of a 2D landscape is optimized to redirect the fluid towards the purple targets; birds eye views of heightmaps are shown in the upper right corner of each subplot. In this high-dimensional task domain, GD finds designs with high reward (a-c), while CEM fails to find meaningful solutions for *2 Pools* (d) as well as the other tasks (Figure A.5a). Each subplot reports the mean reward (r) and bootstrapped [lower, upper] 95% confidence intervals for the corresponding optimizer and task (averaged over 10 randomized initial designs for each task variation, see Section C.2).

curved lines or large numbers of obstacles are shown. Therefore, designs which solve 2D FLUID TOOLS tasks are necessarily out of distribution for the learned model.

**3D WATERCOURSE**    To evaluate higher-dimensional inverse design, we created a "landscaping" task that requires optimizing a 3D surface to guide fluids into different areas of an environment (Figure 3, Section C.2). Specifically, water flows out of a pipe and onto an obstacle parameterized by $\phi_{\mathrm{map}}$, a $25 \times 25$ heightmap (625 design parameters). In *Direction*, $\phi_{\mathrm{map}}$ is optimized to redirect the fluid stream towards a specified direction. In *2 Pools* and *3 Pools*, $\phi_{\mathrm{map}}$ is optimized to split the fluid stream such that particles hitting the floor land as close as possible to one of two or three specified pools. Each task has multiple variants, such as different target directions in *Direction*. The simulation is unrolled for 50 time steps, and contains up to 2048 particles. The learned model $f_M$ is trained on data generated from the simulator in [8].

**AIRFOIL**    Shape optimization in aerodynamics is one area where gradient-based optimization is routinely applied using traditional simulators [13]. Here, we consider the well-studied task of drag optimization of a 2D airfoil profile (Figure 5, Section C.3). In this task, a wing is defined using a curve on a 2D mesh, which can be deformed using a set of 10 control points $\phi_{\mathrm{ctrl}}$. The task is formulated as optimizing the wing shape to minimize drag under certain constraints such as constant lift and bounds on the shape to prevent degenerate (e.g. infinitely thin) configurations. Lift and drag coefficients are computed by running an aerodynamics simulation on a 4158 node mesh. The learned model $f_M$ is trained on data generated with the OpenFOAM solver [55], which pairs wing shapes with targets of RANS pressure and Reynolds stress fields discretized on a 4158 node mesh. Note that unlike the fluids domains, this simulation is *steady-state* (ie. a rollout length of 1).

**Model learning and optimization**    While each domain has a different state space structure and uses a different ground-truth simulator $f_S$, the learned simulators $f_M$ all share the same architecture (with identical hyperparameters in 2D FLUID TOOLS and 3D WATERCOURSE, and only minor variations of the hyperparameters for AIRFOIL due to it being a steady state simulation; see Appendix B). The models are trained for next-step prediction on task-independent datasets (random perturbations of the design space for AIRFOIL and 3D WATERCOURSE, and an open-source, qualitatively distinct dataset for 2D FLUID TOOLS), and are unrolled for up to 300 time steps during design optimization without further fine-tuning (see Section 3 for details). Importantly, all the pre-trained models are shared across the multiple design tasks for each domain.

## 5    Results

Our results show that learned simulators can be used to effectively optimize various designs despite significant domain shift and long rollout lengths. The same underlying model architecture is used for each domain, highlighting the generality of learned simulators for design. Examples of designs found with

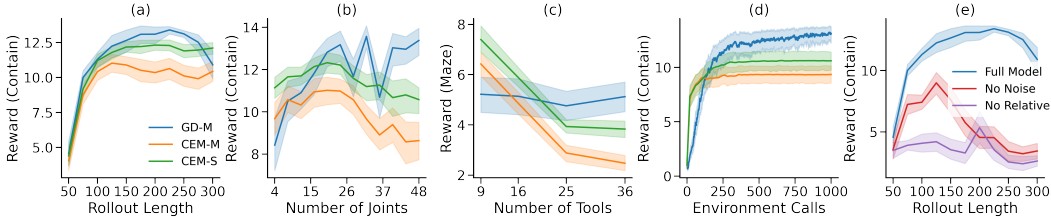

Figure 4: Ablation experiments on the *Contain* and *Maze* tasks. (a) Performance of all optimizers increase with rollout length; GD-M performance starts to deteriorate around step 225. (b) In *Contain*, CEM performance drops when increasing the number of joints above 24, while GD-M remains stable. (c) We observe a similar trend with the number of tools in *Maze*. (d) CEM often gets stuck in sub-optimal solutions early in optimization, while GD-M performance continues to increase. (e) Model choice matters: good performance on GD-M is enabled by key architectural choices of the learned model, such as training noise and translational equivariance.

our approach are available at: `https://sites.google.com/view/optimizing-designs`. Example code can be found at `https://github.com/deepmind/deepmind-research/inverse_design/`. Performance is always evaluated using the ground-truth simulator $f_S$ (see Section 3). Here we discuss these results, and compare the capabilities of gradient descent with learned simulators over classical simulators and sampling-based optimization techniques.

**Overall results**    We first asked whether a learned simulator combined with gradient descent (GD-M) could produce good-quality designs at all. This approach might fail in various ways: accumulating model error, vanishing or exploding gradients [9], or domain shift [36, 71]. However, as the following results show, GD-M produced high-quality designs across all three domains.

Figure 2a shows qualitative results for 2D FLUID TOOLS (Section 4), where our approach (GD-M) produces intuitive, functional designs to contain (*Contain*), transport (*Ramp*), or funnel (*Maze*) the fluid to a target location. On average, GD-M outperforms CEM-M by 16.1–118.9%, indicating a substantial benefit of gradient-based optimization. GD-M also outperforms CEM-S by 3.9–37.5%, despite using a learned simulator rather than the ground-truth. However, these design spaces are still relatively small (between 16 and 36 dimensions). In 3D WATERCOURSE (Section 4), we substantially increase the dimensionality to a 625-dimensional landscape. Here, GD-M produces robust designs, creating ridges to re-route water in particular directions or valleys to direct water into pools (Figure 3a-c). By comparison, CEM-M cannot solve any of these tasks, with performance 30–85× worse than GD-M. In the Appendix (Section D.2) we demonstrate that these results also hold for other optimizer choices (Bayesian Optimization, CMA-ES).

In AIRFOIL (Figure 5b), GD-M recovers the characteristic S-curve shape for a low-drag airfoil under a small angle of attack, and matches the design obtained with DAFoam, an adjoint aerodynamics solver which computes close-to-optimal designs for this task. The design obtained with DAFoam yields a drag coefficient of 0.01902, while GD-M finds designs with drag between 0.01898–0.01919 depending on ensemble size (see Section 5). Importantly, DAFoam's solver and optimizer are highly specialized for the particular task of airfoil design, while our approach is more general-purpose, requiring only trajectory data for training and a generic gradient-based optimizer.

**Model stability & gradient quality**    We investigated accuracy over long timescales by measuring the effect of rollout length on design quality in 2D FLUID TOOLS (Figure 4a and A.11). Longer rollouts can in principle allow for higher reward in this task as they give the fluid time to settle; however, with learned models, they can also be unstable due to error accumulation [69, 72]. Nevertheless, we find that the learned simulator does not seem to be severely impacted by this problem. Specifically, the quality of designs found by GD-M increases up to 225 steps (Figure 4a), indicating that the learned simulator's accuracy and gradients remain stable for a surprisingly long time. Across the episode lengths evaluated, we find that GD-M outperforms not only CEM-M (by 18.1% on average) but also CEM-S (by 4.4% on average). This indicates that the benefits of a having a learned model that supports better optimization techniques can outweigh the small errors incurred by long rollouts.

The strong performance of GD-M on longer rollout lengths is noteworthy. Gradients tend to degrade when passed through chains of many model evaluations, and as a result, previous work generally only optimizes gradients in small action spaces over just a few time-steps [51].

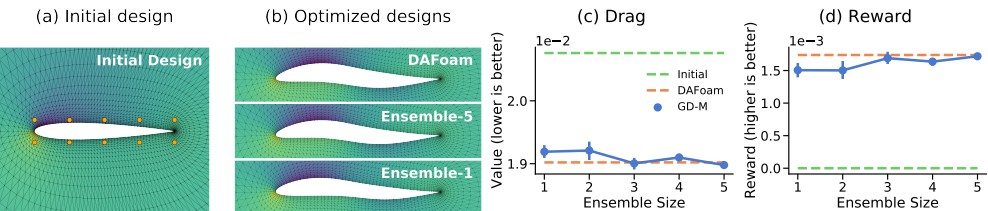

Figure 5: AIRFOIL results. (a) An initial airfoil design is warped by moving 10 control points (orange dots). The physics model simulates the resulting aerodynamics on a 4158 node mesh, based on which lift and drag are computed. (b) For the task of finding a minimum-drag configuration under constant lift constraint, gradient-based learned design is able to find similar designs to specialized solver DAFoam, both using single models and ensembles. (c-d) Larger ensemble sizes of 3–5 achieve a close quantitative match to DAFoam for both drag and overall reward. Shown are means over 10 randomized initial designs, with bootstrapped 95% confidence intervals.

We hypothesize that a key reason for the success of GD-M is the addition of noise in training the model $f_M$, which promotes stability on the forward pass and smoother gradients. We test this by considering a model variant trained without noise (Figure 4e, red). We observe much worse design quality compared to the full model, particularly for longer rollouts.

**Improving accuracy with ensembles**   In engineering tasks like AIRFOIL, simulators must be especially accurate, as small differences in the predicted pressure field can cause large errors in lift and drag coefficients. While GD-M (without ensembling) produces designs close to DAFoam's, we notice a slightly rounder wing front (Figure 5b, bottom) causing a small increase in drag (0.01919 versus 0.01902 in DAFoam).

To further improve performance, we implemented an ensemble of learned simulators trained on separate splits of the training set. Ensembles are a popular choice for training transition models for use in control [20], as they can provide higher quality predictions and are more resistant to delusions—a particularly problematic issue for accuracy-sensitive domains such as airfoil design. During optimization, we make predictions with all models in the ensemble, each trained on a different data split, and average the gradients. As shown in Figure 5b-c, larger ensembles yield designs with significantly lower drag ($\beta = -5.3 \times 10^{-5}$, $p = 0.0003$, where $\beta$ is a linear regression coefficient) and higher overall reward ($\beta = 5.6 \times 10^{-5}$, $p = 0.0001$), and are able to produce designs very close to the solution found by DAFoam, with a drag coefficient of 0.01898 (size-5 ensemble). Thus we are able to achieve performant designs with a general-purpose learned simulator, indicating that we can use learned models for design optimization in spaces traditionally reserved for specialized solvers like DAFoam.

**Scalability to larger design spaces**   In larger design spaces, sampling-based optimization procedures quickly become intractable, especially with relatively slow simulators. We hypothesized that gradient descent with fast, learned simulators could overcome this issue, especially as the size of the design space is increased. We therefore compared different optimizers on 2D FLUID TOOLS as a function of the dimensionality of the design space (the number of tool joints in *Contain* or the number of tools in *Maze*) and on the higher-dimensional 3D WATERCOURSE.

For *Contain* (Figure 4b), the performance of GD-M increases with the number of joints as increasingly fine grained solutions are made possible. In contrast, for both CEM-M and CEM-S, design quality deteriorates with more joints as high quality solutions become harder to find with random sampling. In the highest dimensional *Contain* task with 48 tool joints, GD-M outperforms CEM-M by 154.9% and CEM-S by 126.5%. When CEM does find solutions (Figure 2b), they lack global coherence and appear more jagged than solutions found with GD-M. Similarly, for *Maze* (Figure 4c), the performance of GD-M is largely unaffected by the number of tools, while the performance of CEM-M and CEM-S both degrade as the design space grows. For the highest dimensional *Maze* problem with 36 joints, GD-M outperforms CEM-M by 207.4% and CEM-S by 133.7%.

In the 625-dimensional 3D WATERCOURSE task, CEM-M performs 30–85× worse than GD-M (Figure 3) despite extensive hyperparameter tuning. This trend held across all tasks (Figure A.5a), and even persisted when using fewer control points in the design space (Figure A.7). This is due not only to 3D WATERCOURSE's larger design space, but also because this problem requires a globally coherent solution: small, independent changes have a negligible effect on the global movement of the fluid.

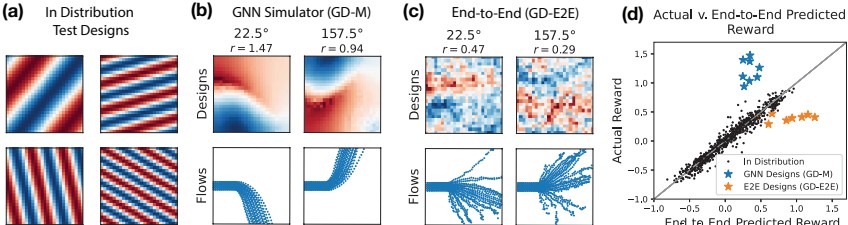

Figure 6: Analysis of an End-to-End surrogate reward model (GD-E2E) versus our approach (GD-M). (a) Test designs sampled from the same distribution as the training data. (b,c) Optimized designs and top view of resulting flow trajectories from the (b) Learned Simulator (GD-M) and (c) End-to-End (GD-E2E) model for two *Direction* tasks in 3D WATERCOURSE. (d) Ground Truth (evaluated using $f_s$) vs Predicted reward for different sets of designs. (black dots) held-out in-distribution test designs, (blue) designs optimized using (GD-M), (orange) designs optimized using (GD-E2E).

**Model speed and sample efficiency**  Learned simulators can provide large speedups over traditional simulators in certain domains by learning to compensate for coarser sub-stepping and making optimal use of hardware acceleration. In AIRFOIL, although we use a very simple GD setup, our approach is able to find very similar designs as DAFoam's specialized optimizer. Moreover, our approach requires only 21s (single model) to 62s (size-5 ensemble) on a single A100 GPU, compared to 1021s for DAFoam run on an 8-core workstation, despite requiring $10\times$ more optimization steps.

In 2D FLUID TOOLS, the ground-truth simulator runs at a similar speed to the learned model [see 63], but is non-differentiable and therefore depends on more expensive gradient-free optimization techniques which require more function evaluations. We use 20–40 function evaluations per optimization step of CEM, compared to a single evaluation with GD (which is about $3\times$ more costly, due to the gradient computation). Thus, GD with a differentiable learned model can be much more efficient than using the ground truth simulator with a sampling-based method.

**Generalization**  Deep networks often struggle to generalize far from their training data [31]. This poses a problem for design: to produce in-distribution training data, we would already need to know what good designs look like, thus defeating the aim of wanting to find *new* designs. However, we find that the GNN-based simulators generalize well in the setting studied here. As noted in Section 4, the learned simulator for 2D FLUID TOOLS was trained on a pre-existing, highly simplified dataset where only one to four straight line segments interact with a fluid (Appendix C). However, it still can support the design tasks studied here, which involve highly articulated, curved obstacles (*Contain*, *Ramp*) or a larger number of obstacles (*Maze*), without further finetuning (Figure 2).

In Section D.6, we show similarly compelling generalization results for 3D WATERCOURSE. Without fine-tuning, we optimize the heightfields of **two** meshes *along with their global rotations* to redirect water. These design tasks now consist of 1250 + 2 design dimensions, while also being further from the training distribution (only *one, un-rotated* heightfield was ever observed). Despite these added challenges, the method still produces good designs with reward between 0.52 and 1.16.

This generalization capability emerges due to certain design choices of the simulator model $f_M$. Specifically, the GNN operates on local neighborhoods of particles, and particle coordinates are represented in a relative manner, which encourages learning local rules that can transfer to very different setups. We tested a model variant which breaks this property by encoding *absolute*, instead of *relative* particles coordinates (Figure 4e, purple). We find that this variant can no longer robustly simulate out-of-domain designs, leading to worse performance at any rollout length.

Using a learned simulator trained in a relatively simple environment has another unexpected advantage. In rare cases, classical simulators suffer from degenerate behavior around certain edge cases. For example, with the classical simulator for 2D FLUID TOOLS, particles can get stuck in between joint segments, especially when there are a large number of parts or joints (Figure A.10). However, since the learned simulator was trained on simpler data where these effects are unobserved, it picks up only on the appropriate collision performance and not the unrealistic edge cases. Thus, the learned simulator produces more plausible rollouts than the classical simulator in these cases, and might therefore be a better candidate for producing designs that would transfer to the real world. There are also cases where we would expect the classical simulator to generalize better than the learned simulator, such as in dealing with fluids with viscosities outside the training data with different local dynamics.

**Comparing to end-to-end surrogate models**    To further benchmark the generalization capabilities of our approach, we compare to an end-to-end surrogate model trained to map design parameters ($\phi_{\mathrm{map}}$) directly to rewards for each of the $\mathbf{d} = 8$ target angles of the *Direction* task in 3D WATER-COURSE. To train the end-to-end surrogate model, we use the same designs as those used to train the GNN dynamics model (Section C.2), but additionally paired with the rewards predicted by the ground truth simulator, so that $\mathcal{D}_{sur} = \{(\phi_{\mathrm{map}}, f_{R_{\mathrm{dir}}}(\phi_{\mathrm{map}}, \mathbf{d}))^{(i)}\}_{i=0}^{1000}$. We use a ResNet-18 [39] as the surrogate model with a $25 \times 25$ dimensional input image corresponding to the heightfield mesh, and we train a separate model for each target direction. We focus on 3D WATERCOURSE for these experiments, as the training data and design problems contain meshes in the same domain ($25 \times 25$ meshes). For 2D FLUID TOOLS the training data and design spaces are different, making the end-to-end surrogate approach fundamentally under-determined.

The surrogate model could fit the training set and an in-distribution test set well (MSE=$0.0070 \pm 0.0025$, Figure 6d). However, when used for design optimization with a GD optimizer, the surrogate moves into out of distribution regions of design space where it makes overly optimistic predictions (predicted=$1.12 \pm 0.31$, actual=$0.40 \pm 0.05$, MSE=$0.615 \pm 0.403$, Figure 6b,c). The surrogate model also severely underestimates the reward of high-scoring designs found using our method GD-M (predicted=$0.33 \pm 0.06$, actual=$1.21 \pm 0.18$, MSE=$0.813 \pm 0.344$, Figure 6d). Overall, this suggests that using task-agnostic, dynamics-based models for design can have appealing regularization properties which avoid issues of model delusion that are commonly seen with end-to-end surrogate approaches.

# 6   Discussion

We find that state-of-the-art learned, differentiable physics simulators can be used with gradient-based optimization to solve challenging inverse design problems. Across three domains and seven tasks, which involved designing landscapes and tools to control water flows or optimizing the shape of an airfoil, we demonstrated that gradient descent with pre-trained simulators can discover high-quality designs that match or exceed the quality of those found using alternative methods. This approach succeeds in a variety of interesting and surprising ways: it permits gradient backpropagation through complex physical trajectories for hundreds of steps; scales to tasks with large design and state spaces (100s and 1000s of dimensions, respectively); and successfully generates designs which require the learned simulator to generalize far beyond its training data. In the classic aerodynamics problem of airfoil shape optimization, our approach produces a design comparable to that of a specialized solver using only simple, general-purpose strategies like model ensembling.

While our results have exciting implications for inverse design, they also open up possibilities for explaining everyday human behavior like tool invention—a longstanding puzzle in cognitive science [1, 56, 67]. With general-purpose learned simulators, we have the potential not just to create highly specialized tools in engineering domains, but also to model everyday tool creation, such as creating a hook from a pipe cleaner, building a blanket fort, or folding a paper boat.

Our approach has limitations that should be visited in future work. Gradient descent is inappropriate for design spaces with regions of zero gradients, such as in 2D FLUID TOOLS tasks where the fluid may not always make contact with the tool (Figure A.9). Many interesting design tasks also have variably-sized or combinatorial design spaces that cannot easily be optimized with gradient descent, such as computer-aided design (CAD) approaches to 3D modeling. An exciting future direction will be to integrate general-purpose learned simulators with hybrid optimization techniques such as those used in material science and robotics [17, 70]. As learned simulators continue to improve, we could also use them to do even broader cross-domain, multi-physics design. While challenges remain, our results represent a promising step towards faster and more general-purpose inverse design.

# 7   Acknowledgements

We thank Yulia Rubanova and Evan Shelhamer for their detailed comments on the manuscript, and Yusuf Aytar, Jonathan Godwin, Luis Piloto, Remi Lam, and Meire Fortunato for helpful discussions.

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
