# A  Optimizer hyperparameters

Optimization hyperparameters were chosen to reflect good performance for each optimizer in each domain. We therefore performed sweeps for major hyperparameters of each optimizer for each domain, with those used for experiments in the paper shown in the table below.

CEM maintains a population of samples and uses these to estimate the mean $\mu$ and standard deviation $\sigma$ of a Gaussian distribution over design parameters. To optimize $\mu$ and $\sigma$, it takes the top performing fraction, deemed the "elite portion," from the current step. The initial standard deviation of this distribution is given by "Initial $\sigma$", and the initial mean is set to 0. We found that for CEM, the population sample size had a significant effect on overall optimization quality (Figure A.1). Due to computational considerations, we picked the smallest value for this hyperparameter that performed within 1 standard deviation of the optimal sample size. Both the elite portion and initial $\sigma$ parameters were chosen as the best performing values on a set of held-out random tasks for each domain.

For GD, we only performed a hyperparameter sweep over the learning rate, which was the only parameter to significantly affect performance. For the 2D FLUID TOOLS tasks, we introduced gradient clipping to eliminate the effect of rare gradient spikes over the course of optimization. However, not using gradient clipping still produced qualitatively and quantitatively similar results. For additional Adam parameters, we used the default values for the exponential decay rates that track the first and second moment of past gradients of $b_1 = 0.9$ and $b_2 = 0.999$ [41].

|  | 2D FLUID TOOLS | | | 3D WATERCOURSE | | AIRFOIL |
| --- | --- | --- | --- | --- | --- | --- |
| GD | *Contain* | *Ramp* | *Maze* | *Direction* | *Pools* | |
| Learning rate | 0.005 | 0.005 | 0.01 | 0.01 | 0.01 | 0.01 |
| Momentum term $b_1$ | 0.9 | 0.9 | 0.9 | 0.9 | 0.9 | 0.9 |
| Momentum term $b_2$ | 0.999 | 0.999 | 0.999 | 0.999 | 0.999 | 0.999 |
| Gradient clip | 10 | 10 | 10 | — | — | — |
| CEM | | | | | | |
| Sampling size | 20 | 20 | 20 | 40 | 40 | — |
| Elite portion | 0.1 | 0.1 | 0.1 | 0.1 | 0.1 | — |
| Initial $\mu$ | 0 | 0 | 0 | 0 | 0 | — |
| Initial $\sigma$ | 0.5 | 0.5 | 1.5 | 0.1 | 0.1 | — |
| Evolution smoothing | 0.1 | 0.1 | 0.1 | 0.1 | 0.1 | — |
| Optimization steps | 1000 | 1000 | 1000 | 200 | 200 | 200 |

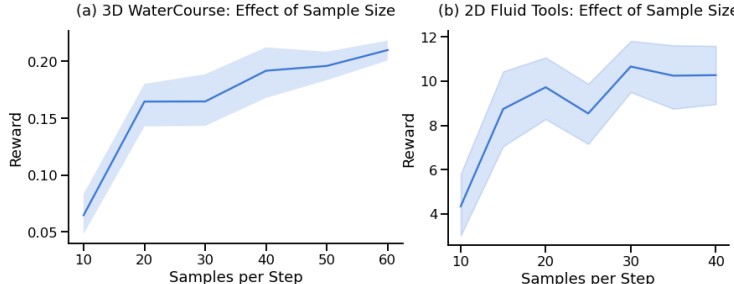

Figure A.1: For CEM, increasing population size, while more computationally expensive, can lead to improvements in performance. (a) In the 3D WATERCOURSE domain, CEM benefits from large sample sizes, although returns are diminishing for sizes beyond 40 (*Direction* task, 36 design parameters). (b) In the 2D FLUID TOOLS domain, CEM benefits from larger sample sizes, although returns are diminishing for sizes beyond 20 (*Contain* task, 40 design parameters).

# B    Model architecture and training

For each task domain, we train a GNN for next-step prediction of the system state. For the domains considered in this paper, we unify the approaches of GNS [63] and MESHGRAPHNETS [57]: In the AIRFOIL domain, we encode/decode mesh nodes and mesh edges as a graph as described in the aerodynamics examples of MESHGRAPHNETS, while for particle-based fluids, edges are generated based on proximity as in GNS. In the case of 3D WATERCOURSE, both particles (fluid) and a mesh (the designed obstacle) are present; hence, edges are generated based on proximity (for fluid-fluid and fluid-obstacle interaction) or from the landscape mesh. As the landscape does not have any internal dynamics, we did not find it necessary to distinguish between world- and mesh edges, and use a single edge type.

Once encoded as a graph, the core model and training procedure is largely identical between GNS and MESHGRAPHNETS, and we refer to the above papers for full details on architecture and model training. Briefly, we use an Encode-Process-Decode GNN with 10 processor blocks. All edge and node functions are 2-layer MLPs of width 128, with ReLu activation and LayerNorm after each MLP block. The model is trained with Adam and a mini-batch size of 2, with training noise, for up to 10M steps. We implemented this model in JAX [11]. In addition to the different encoding procedures for mesh vs. particle systems, the parameters for training noise and connectivity radius have to be set per-domain, to account for differences in particle size/mesh spacing. These details are described in Appendix C. The models were trained on 4 TPUv3 for 4-10 days.

**Gradient computation**    In our experiments, we pass gradients through long model rollouts of up to 300 steps. As it is prohibitive to store all forward activations for the backwards pass, we use gradient checkpointing [18] to store activations only at the beginning of each step of the trajectory during the forward pass, and recompute the intermediate activations for each step as needed when the backwards pass walks the trajectory in reverse. Gradient calculation using this method has roughly 3 times the time cost of a pure forward simulation: forward dynamics have to be computed twice for each step, in addition to the computation of the backwards pass itself.

# C    Task domains

## C.1    2D FLUID TOOLS

Tasks in 2D FLUID TOOLS are procedurally generated from templates specified in Table A.1. The simulation domain is a 2D box, with the lower left corner specified as $[0, 0]$, and upper right corner specified as $[1, 1]$. Fluid particles are initialized as a box of size *Initial fluid box* with bounding boxes given in format $[x_{\min}, y_{\min}, x_{\max}, y_{\max}]$. Certain task parameters were varied for ablation experiments in Figure 4 (rollout length, # joints (*Contain*), # tools (*Maze*)); Table A.1 contains default values used unless otherwise specified.

**Design space**    A "tool" in this task domain is a 2D curve composed of several line segments connected by joints. For a large number of joints, a tool can thus approximate a smooth curve (Figure A.2). Each task's design space consists of the relative joint angles controlling the tool's shape. We consider tasks with a single, multi-segment tool (*Contain*, *Ramp*) and a task with multiple, single-segment tools (*Maze*). For each tool, relative angles are calculated by moving from the anchor point on the left, along the tool segments to right, such that $\text{angle}_i = \text{angle}_{i-1} + \phi_{\text{joints}_i}$ for the $i^{\text{th}}$ joint from the anchor. We also experimented with two additional design space parameterizations: (1) jointly optimizing the joint angles and a global position offset $[x, y]$ for each tool, and (2) changing the parameterization of angles to be absolute (such that $\text{angle}_i = \phi_{\text{joints}_i}$ directly). We discuss the effects of these alternate parameterizations in Section D.3.

**Simulation and objective**    Both fluids and tools are represented as particles with different types, and simulated with the learned model for 150 steps (with the exception of the ablation experiment on rollout length). Scenes consist of $N = 100 \ldots 1000$ fluid particles. For ground-truth evaluation of the designs, we simulate particle dynamics with an MPM solver [43]. Task reward is calculated using the Gaussian likelihood of the final particle positions after rollout ($\mathbf{u_v}$ from $\tilde{G}^{t_K}$). That is, for a task with reward parameterized with mean $\mu$ and spherical covariance $\sigma$ ($\theta_R = [\mu, \sigma]$), the reward is

| | Contain | Ramp | Maze (nxn) |
|---|---|---|---|
| Environment size | 1x1 | 1x1 | 1x1 |
| Rollout length | 150 | 150 | 150 |
| Initial fluid box | [0.2, 0.5, 0.3, 0.6] | [0.2, 0.5, 0.3, 0.6] | [0.2, 0.75, 0.8, 0.8] |
| Reward sampling box | [0.4, 0.1, 0.6, 0.3] | [0.8, 0, 1, 0.2] | [0.1, 0.1, 0.9, 0.2] |
| Reward $\sigma$ | 0.1 | 0.1 | 0.1 |
| Design parameter | joint angles | joint angles | rotation |
| # tools | 1 | 1 | $n^2$ |
| # joint angles | 16 | 16 | 1 |
| Tool position (left) | [0.15, 0.35] | [0.15, 0.35] | — |
| Tool domain box (3x3) | — | — | [0.14, 0.3, 0.65, 0.6] |
| Tool domain box (4x4) | — | — | [0.14, 0.3, 0.71, 0.6] |
| Tool domain box (5x5) | — | — | [0.14, 0.3, 0.75, 0.6] |
| Tool domain box (6x6) | — | — | [0.14, 0.25, 0.77, 0.65] |
| Tool Length | 0.8 | 0.8 | — |
| Tool length (3x3) | — | — | 0.72 |
| Tool length (4x4) | — | — | 0.64 |
| Tool length (5x5) | — | — | 0.65 |
| Tool length (6x6) | — | — | 0.63 |

Table A.1: Task Parameters for 2D FLUID TOOLS tasks. Boxes are described as $[x_{\min}, y_{\min}, x_{\max}, y_{\max}]$.

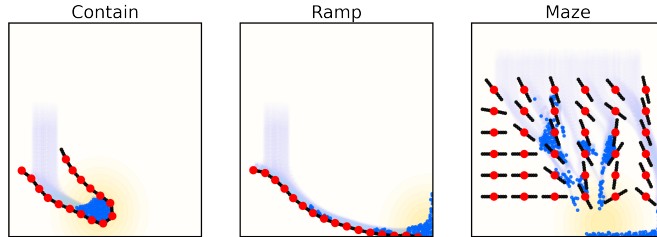

Figure A.2: Visualization of the design space parameterization for the 2D FLUID TOOLS task. Each red dot corresponds to the anchor points (*Contain* and *Ramp*) and center of rotation (*Maze*) being optimized.

calculated as

$$f_R := \text{mean}_v \, \log(\mathcal{N}(\mathbf{u_v}; \mu, \sigma)) \,.$$

***Contain*** For this task, the center of the goal region $\mu$ is sampled uniformly from a rectangular reward region in the lower-middle section of the $1 \times 1$ simulation domain ($[0.4, 0.6] \times [0.2, 0.4]$). A tool protruding to the right is initially placed below the fluid rectangle. By optimizing a single tool's relative joint angles, successful solutions must "contain" the fluid in the region by creating a cup or spoon.

***Ramp*** The fluid and tool are initialized as in *Contain*, and $\mu$ is sampled from a region lower and further to the right than in *Contain* ($[0.8, 1] \times [0, 0.2]$). By again optimizing a single tool's relative joint angles, successful solutions will create a "ramp" from the initial fluid position to the goal location in the bottom right.

***Maze*** The goal is sampled from a long region near the bottom of the domain ($[0.1, 0.9] \times [0.1, 0.2]$). By optimizing the rotation angles of a grid of rigid, linear tools, successful solutions will create a directed path from the top of the screen to the goal location at the bottom.

**Model training** We trained the learned simulator on the WATERRAMPS datasets released by Sanchez-Gonzalez et al. [63]. This dataset consists of 1000 trajectories featuring a single large block of water falling on one to four randomized straight line segments (see image below for examples). Model architecture and hyperparameters are described in Appendix B, with a training noise scale of $6.7\,10^{-4}$ and connectivity radius of $0.015$.

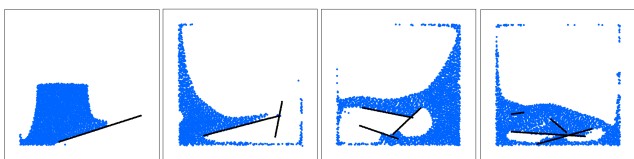

Figure A.3: Four examples of trajectories from the WATERRAMPS dataset released by [63] used as training data for the supervised prediction model.

## C.2 3D WATERCOURSE

**Design space** This domain has a design space $\phi_{\mathrm{map}}$ of 625 parameters, which determine the y coordinate offset to nodes of a $25 \times 25$ square mesh centered at $\mathbf{c} = (0.3, 0.5, 0.5)$ in the simulation domain. While we could directly mapping the parameters to coordinates, we use the design function $y_i = \gamma_H \tanh(\phi_{\mathrm{map}_i})$ ($\gamma_H = 0.3$ for all tasks) to prevent trivial task solutions (i.e. obstacles which touch the floor).

**Simulation** The simulation consists of an inflow pipe located at (-0.5, 1.0, 0.5) above the landscape which continually emits a stream of liquid, represented as particles. These particles are then redirected by the designed landscape, and finally removed once they hit the floor at $y = 0$. In our experiments we observed up to 2084 particles present in the scene at one time. We unroll the learned simulation model for trajectories of 50 time steps, and store the final particle positions $\mathbf{u_v}$, as well as the positions of removed particles that touched the floor at any point $\mathbf{u_v^D}$ to be passed to the reward function. Ground truth simulations for evaluation are performed by running the same setup with an SPH solver. We note that SPH requires very small simulation time steps, and performs $\approx 10^4$ internal steps for a trajectory of the same length.

***Direction*** In this task, we want to align the water stream with a given direction vector $\mathbf{d}$. We can formalize this using the reward function

$$f_{R_{\mathrm{dir}}} := \mathrm{mean}_v\left((\mathbf{u_v} - \mathbf{c}) \cdot \mathbf{d}\right) - \mathrm{std}_v\left((\mathbf{u_v} - \mathbf{c}) \cdot \mathbf{d}_\perp\right) - \gamma_R \, \mathrm{mean}(\nabla \phi_{\mathrm{map}})$$

where $\mathbf{d}_\perp$ is orthogonal to $\mathbf{d}$. The first term aligns the direction of the particle relative to the domain center, and the second term concentrates the stream. The last term is a smoothness regularizer on the design landscape, which prefers smooth solutions ($\gamma_R = 300$ for both tasks). Absolute reward numbers for this task can be positive or negative, hence we report the normalized reward $f_{R_{\mathrm{dir}}} - f_{R_{\mathrm{dir}}}^{\mathrm{initial}}$, i.e. an unchanged initial design corresponds to a zero reward, to make the scores easier to interpret.

Rewards can be in 8 different directions, spaced between $0$ and $180\,\mathrm{deg}$. We collapse across directions for reporting reward means and confidence intervals for each optimizer.

***2 Pools*** and ***3 Pools*** In these tasks, we define two and three pools, respectively, with center $\mu_\mathbf{p}$ on the floor. For each particle which has hit the floor, we assign it to its closest pool $\hat{\mu}_\mathbf{p}$, and define the reward as the Gaussian probability under $\hat{\mu}_\mathbf{p}$, i.e.

$$f_{R_{\mathrm{pools}}} := \mathrm{mean}_v(\mathcal{N}(\mathbf{u_v^D}; \hat{\mu}_\mathbf{p}, \sigma)) - \gamma_R \, \mathrm{mean}(\nabla \phi_{\mathrm{map}})$$

with $\sigma = 0.4$ and a regularization term as above.

To showcase different ways of splitting the water stream, we consider one positioning of the pools for the two pool case, and two for the three pool case. In the two pool case, pools are placed at $[1.49, -0.35]$ and $[1.49, 1.35]$. In the three pool case, pools are placed either at $[1.6, -0.45]$, $[1.85, 0.5]$, and $[1.6, 1.45]$, or at $[0.5, -0.5]$, $[1.7, 0.5]$, and $[0.55, 1.5]$. These were selected to ensure the task was solveable – pools directly beneath the landscape, or too far away from the landscape, would not be reachable even with dramatically warped surfaces.

**Model training**   We trained a model on next-step prediction of particle positions, on a dataset of 1000 trajectories of water particles interacting with a randomized obstacle plane (random rotations and sine-wave deformations of the planar obstacle surface). The data was generated using the SPH simulator SPlisHSPlasH [8]. The noise scale is set to $0.003$ and a connectivity radius of $0.01$ to account for the different particle radius of the 3D SPH simulation compared to 2D MPM. All other architectural and hyperparameters are as described in Appendix B.

### C.3   AIRFOIL

The airfoil optimization task is modeled similarly to the NACA0012 aerodynamic shape optimization configuration for incompressible flow for the DAFoam solver (see details here), to make it easier to compare design solutions to this solver.

**Design space**   The design space consists of the $y$-coordinate of 10 control points (see Figure 5a). Moving these control points deforms both the airfoil, and the simulation mesh surrounding it. The airfoil shape is deformed using B-spline interpolation as described by Reid [58], and the mesh is deformed using IDWarp [65]. We thus define a design function $G^{t0} = f_D(\phi_{\mathrm{ctrl}}, G_\alpha)$ which takes an initial, undeformed airfoil mesh (we use the standard NACA0012 airfoil), encoded as a graph $G_\alpha$, as well as the control point position $\phi_{\mathrm{ctrl}}$ as input. It returns the graph of the deformed airfoil mesh $G^{t_0}$ to be passed to the simulator. We note that the coefficients for spline interpolation and mesh warping can be precomputed for a given initial mesh, making it easy to define a differentiable function to use for design optimization.

**Simulation**   Given the initial mesh, as well as simulation parameters, the simulator or learned model predict the steady-state incompressible airflow around the wing, sampled on each of the 4158 nodes on the simulation mesh. The entire simulation domain and an example prediction of the pressure field are shown in Figure A.4a,b. For drag minimization we require predictions of the pressure field $p$, as well as the effective Reynolds stress $\rho_{\mathrm{eff}}$ at each mesh node, i.e. $\mathbf{q}_v = (p, \rho_{eff})$. Unlike the other domains in this paper, this is a single-step prediction task, and model rollouts are of length one. For this task, we consider an inflow speed of 0.1 mach, under an $5.1°$ angle of attack.

**Task objective**   The task reward is defined as $f_R := -C_D - \gamma_L||C_L - C_{L0}||^2 - \gamma_A a(\phi_{\mathrm{ctrl}})$, i.e. we minimize the drag coefficient $C_D$ under soft constraints of unchanged lift $C_L$ and a wing area $a$ of 1-3 times the initial area. We use $\gamma_L = 10, \gamma_A = 1$, and a tanh nonlinearity to enforce the volume inequality. Lift and drag can be computed from the simulation output $p, \rho_{eff}$ by integration around the airfoil, see e.g. [49]. We report the normalized reward $f_R - f_R^{\mathrm{initial}}$ such that the initial, undeformed wing design corresponds to a zero reward.

**Model training**   We trained a model to predict $p, \rho_{eff}$ on a dataset of 10000 randomized airfoil meshes, simulated with OpenFoam [55]. For training ensemble models, this dataset is split into 5 non-overlapping blocks, and a separate model is trained on each section. Since this is a steady-state prediction task, information needs to propagate further at each model evaluation. We therefore use twice-repeated processor blocks with shared parameters, i.e. the model performs 20 message passing steps, with 10 blocks of learnable parameters. We found that this increases accuracy in the one-step setup by being able to pass messages further across the mesh. Training noise is often cited for stability over long rollouts, but even in this one-step setting, training noise and data variation can be useful. To increase robustness to unseen wing configurations, we varied the grid resolution between 1000-10000 nodes for each sample in the training set, and added training noise to the input mesh coordinates. We use a normal noise distribution with the scale of $1\%$ of the average edge lengths surrounding the node noise is applied to. All other aspects of model architecture and training procedure are as described in Appendix B.

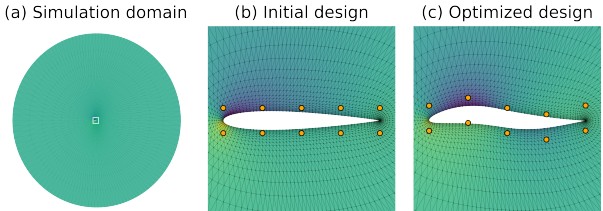

Figure A.4: (a) Aerodynamics are computed on a large 4158 node mesh centered around the airfoil, with the closeup regions around the airfoil in (b, c) marked as a white square in the center. (b, c) Pressure predictions and control points (orange) for the initial and final optimized wing design (Ensemble-5 model).

# D    Further results

## D.1    Model accuracy

In order for a learned simulator to be useful for design, it must be sufficiently accurate in the forward direction. We study this question directly for each of the domains (3D WATERCOURSE, 2D FLUID TOOLS and AIRFOIL) by examining the magnitude of the error between the model predictions of reward for the discovered designs, and the ground truth reward for those designs. The results for each domain are shown in Figure A.5.

Broadly, the learned model very successfully mimics the ground truth simulator in reward prediction across all three domains. The accuracy for AIRFOIL is within a single standard deviation across all ensemble sizes, while the predictions in 3D WATERCOURSE match very closely for both the high performing designs (GD-M) and low performing designs (CEM-M).

However, we do notice some discrepancies in the predicted and ground truth reward for the 2D FLUID TOOLS domain, particularly the *Maze* task. As mentioned in the main text, the ground truth solver sometimes produces unrealistic rollouts for this domain (see Figure A.10), with fluid particles becoming stuck between the different tools. Despite this issue, we find that the model is sufficiently similar to the ground truth to produce designs that still achieve high reward overall.

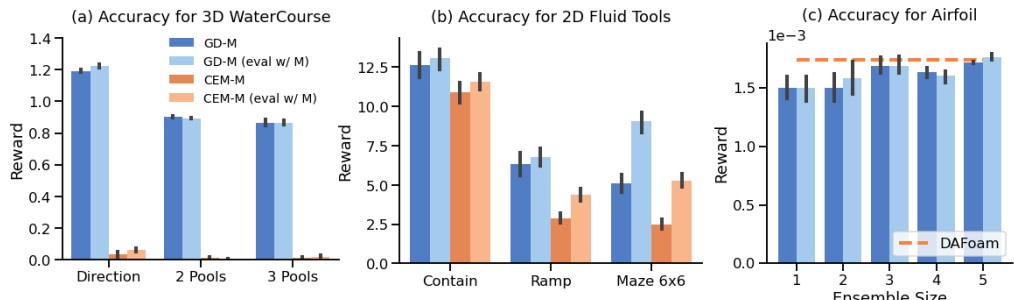

Figure A.5: (a) For the 3D WATERCOURSE domain, reward predicted by the learned model (GD-M eval w/ M, CEM-M eval w/ M) is very close to the ground-truth simulator evaluation (GD-M, CEM-M) for all tasks. (b) This is also true for the 2D FLUID TOOLS domain, though the reward is slightly overestimated when using the model. This effect is amplified in *Maze*, where the ground-truth dynamics sometimes struggles to correctly simulate "sticky" bottlenecks (see Figure A.10). (c) Model predictions of drag (GD-M eval w/ M) are relatively close to the ground-truth simulator evaluation (GD-M), particularly for larger ensemble sizes.

## D.2    Other optimizers

In our experiments in Section 5, we chose the cross-entropy method (CEM) as a popular optimizer representative of model predictive control methods [26, 20], but there are several other possible choices of optimizer. In particular, here we provide results from optimizers that represent two

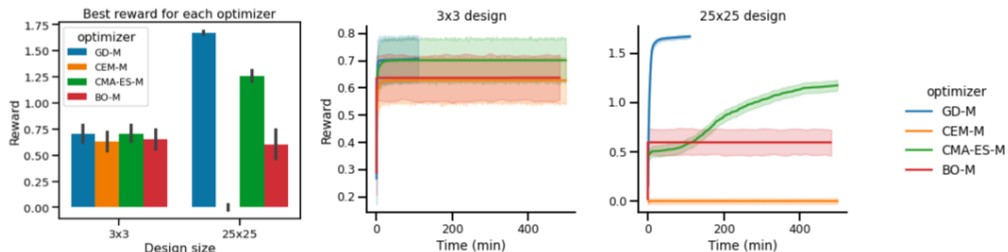

Figure A.6: Performance of alternative optimizers as a function of optimization time. Left: overall performance for each optimizer on the 3D Directions task. Center/Right: performance with a design space consisting of a 3x3 and 25x25 grid, respectively. For large design spaces, GD-M converges significantly faster than alternative optimization schemes, and leads to higher rewards. Error bars and shaded regions represent 95% confidence intervals over 9 directions and 10 seeds per direction.

alternative classes: bayesian optimization [68] (BO, which uses active sampling to explore the design space more effectively), and CMA-ES [38] (which is an evolutionary search method and represents a more sophisticated sampling technique than CEM).

We used the Bayesian Optimization implementation from [33] with their automatic hyperparameter tuning that uses automatic relevance determination (ADR) [74] with a Matérn kernel and the expected improvement (EI) acquisition function. For CMA-ES we swept across the $top_k$ to keep, the smoothing parameter across iterations, the number of samples to use in each step, and the initial standard deviation of the sampling distribution. These were fit on the direction template with a single direction across 2 seeds per hyperparameter. The best performing hyperparameters for CMA-ES were to keep the top 10% of samples, no smoothing across optimization iterations, 40 samples per optimization step, and an initial standard deviation of 0.025. We show results for the 3D WATERCOURSE Direction template to showcase the scalability of both methods at the 3x3 design space and 25x25 design space resolutions.

For a low-dimensional design space (3x3), all optimizers perform comparably. While CMA-ES and GD slightly outperform CEM and BO, the differences are minor. For larger design space sizes (25x25), GD-M significantly outperforms all alternative optimizers. Given enough optimization time, CMA-ES is able to achieve about three quarters of the reward obtained by GD, although it takes roughly 50× as long to do so, as, like CEM, it needs to sample a significantly larger number of environment calls for each optimization step. It outperforms CEM because it represents a full covariance matrix across the top samples in the population, rather than a single scalar.

## D.3    Effects of design parameterization

In this section, we study how different parameterization choices for the design space affect both gradient-based and sampling-based optimizers.

First, in the 3D WATERCOURSE domain, we investigate a parameterization of the design space that uses interpolation to minimize the number of control points on the 2D heightfield (Figure A.7). Control points are placed evenly across the grid, and bi-linearly interpolated onto the $25 \times 25$ mesh. We vary the number of control points from $2 \times 2$ up to $14 \times 14$, and find that while CEM-M performs similarly to GD-M when very few control points are allowed, its performance quickly drops as more control points are added.

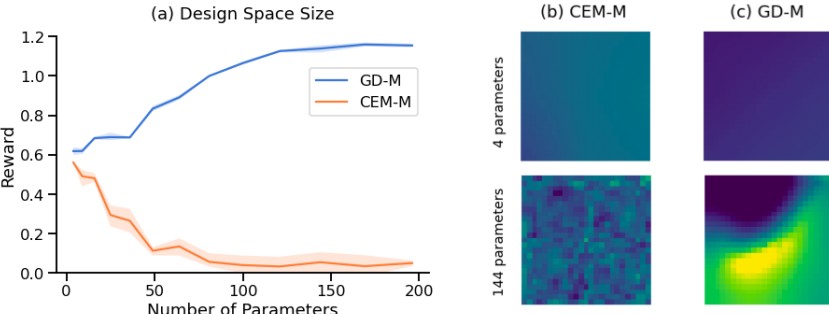

Figure A.7: Performance on the 3D WATERCOURSE *Direction* task with variable design space resolution: GD performs well for large design spaces, while CEM performance quickly drops with increased number of design parameters. (b) Examples of CEM designs at two different design space resolutions. (C) Examples of GD designs at two different design space resolutions.

Second, in the 2D FLUID TOOLS domain, we investigate what happens when we change the design space to use *absolute* joint angles rather than relative ones. When using relative joint angles, changes to joints near the tool's pivot (left side) affect the global properties of the tool. We hypothesized that this could be selectively benefiting the sampling-based approaches, as this makes the effective design space much lower dimensional. We therefore change the design space to be absolute, with a tool's joint angles calculated directly: $angle_i = \phi_i$.

As hypothesized, this change does dramatically decrease the performance of the sampling-based technique (see Figure A.8). Perhaps more surprisingly, the gradient-based optimizer is almost completely unaffected by this reparameterization. While the qualitative solutions it finds differ (with tools now containing "kinks" to prevent the motion of the fluid rather than curves, Figure A.8 top), the overall reward achieved is similar.

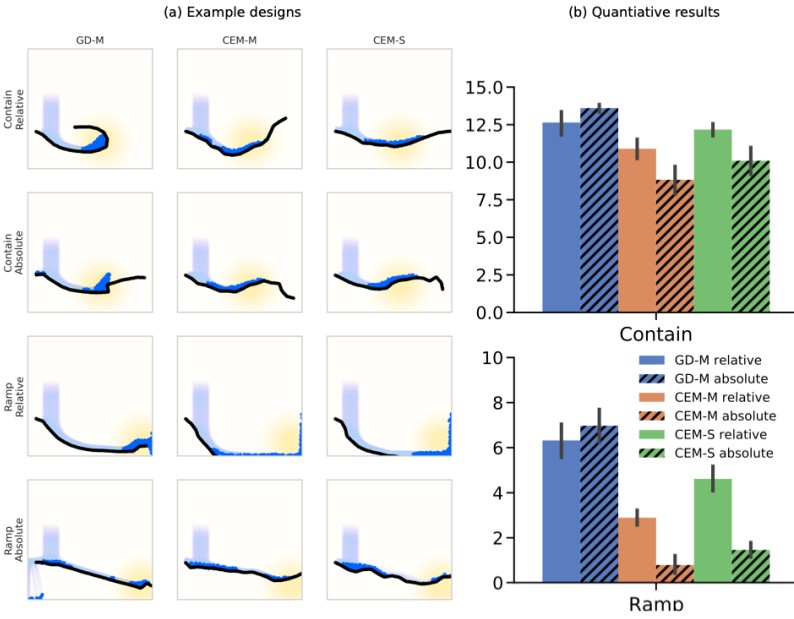

Figure A.8: **(a)** Example solutions for each optimizer across the *Contain* and *Ramp* tasks when optimizing over *Relative* vs *Absolute* angles. **(b)** Mean reward with $95\%$ confidence intervals obtained by each optimizer across the *Contain* and *Ramp* tasks when optimizing over *Relative* vs *Absolute* angles.

### D.3.1 Failure modes of gradient descent

Other parameterizations of the design space can badly affect the performance of gradient-based optimizers. In particular, gradient-based optimizers suffer when there are regions of zero gradients. In the AIRFOIL and 3D WATERCOURSE domains, this is not normally a problem, as the design always interacts with the physical system on which reward is being measured. But in the 2D FLUID TOOLS domain, we can manipulate this.

In particular, for this experiment we changed the design space for 2D FLUID TOOLS to include a global position offset $[x, y]$ for the tool. Making this simple change often has no effect on the discovered designs, but occasionally the gradient-based optimization procedure can move the tool such that it no longer interacts with the fluid (Figure A.9). Once the tool has been moved out of the range of the fluid, there is no longer any way to affect the reward, and therefore there is no gradient signal to recover. To overcome this problem, future work would need to consider more sophisticated hybrid optimization techniques [70].

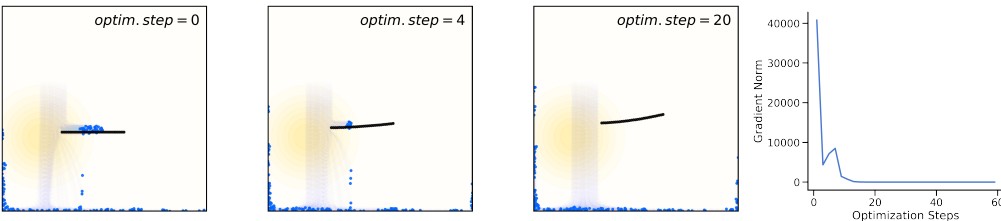

Figure A.9: Failure mode of the GD optimizer: in some instances where the translation of the tool is included in the design, the tool may end up outside of the scope of the fluid. In this cases, the optimization can no longer recover as it will get zero gradients from there on.

## D.4 Failure modes of the MPM solver

One of the advantages of using a learned simulator over a classic simulator is learned simulators can be trained in regions of the state and action space that are known to exhibit regularized, smooth behaviors. For example, as mentioned in Section D.1, the MPM solver [43] we use for evaluation in 2D FLUID TOOLS shows surprising irregularities with "sticking" behavior when there are a large number of different tools. In the *Maze* task, this is particularly prevalent, as fluids often become stuck stochastically in some funnels but not others of similar sizes (Figure A.10).

Since the learned simulator was trained on much simpler scenarios where this effect is not observed, it only learns the smooth behavior of the fluid's movement, which makes the resulting trajectories look more realistic. This may enable better generalization to real world scenarios.

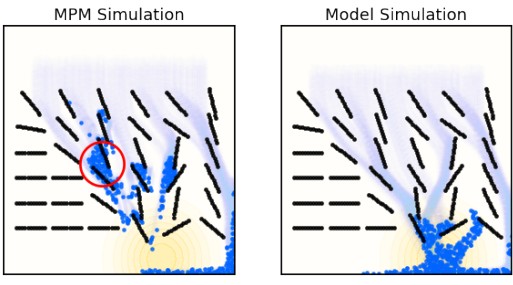

Figure A.10: Left: MPM simulation [43] of a problem with many separate solid objects. As highlighted in the red circle, the MPM solver struggles with water movement between obstacles, often creating artificially sticky bottlenecks. Right: Learned model rollout for the same setup. The model rollout looks significantly more plausible, without any "stickiness" artifacts. Please see https://sites.google.com/view/optimizing-designs for videos demonstrating this effect clearly.

### D.4.1 Further designs found in 2D FLUID TOOLS

In the figures below, we demonstrate the range of found solutions for different solvers in tasks in the 2D FLUID TOOLS domain.

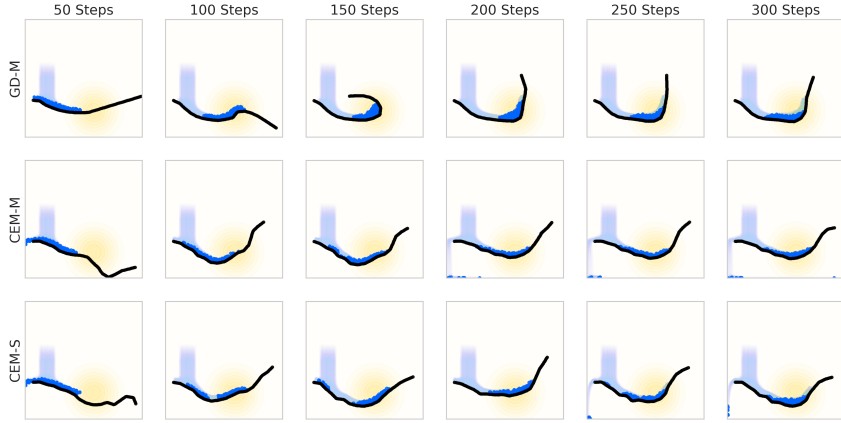

Figure A.11: Example solutions for each optimizer across the range of rollout lengths sampled for *Contain* in Figure 4.

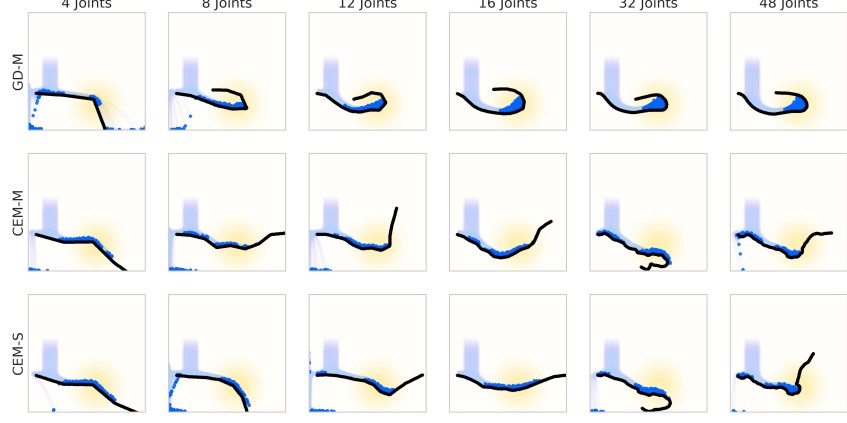

Figure A.12: Example solutions for each optimizer across the range of joint angle numbers sampled for *Contain* in Figure 4.

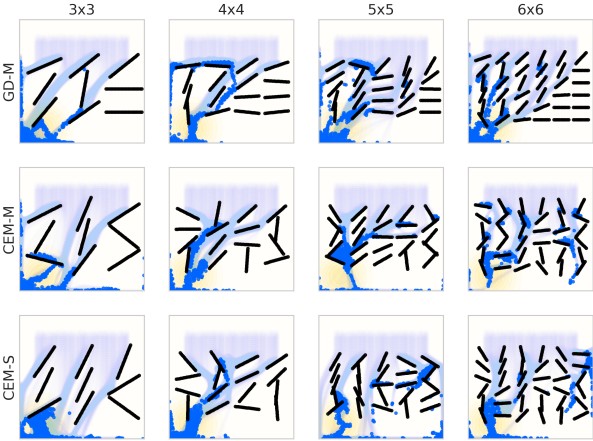

Figure A.13: Example solutions for each optimizer across the range of grid sizes sampled for *Maze* in Figure 4.

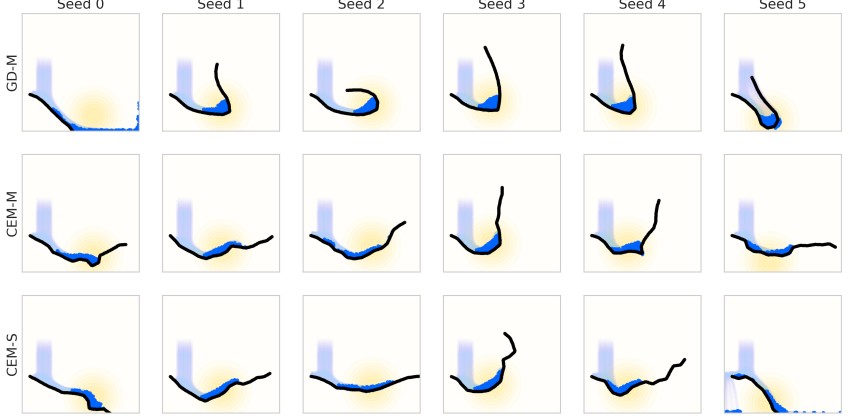

Figure A.14: Example solutions on *Contain* task for each optimizer across 6 random seeds.

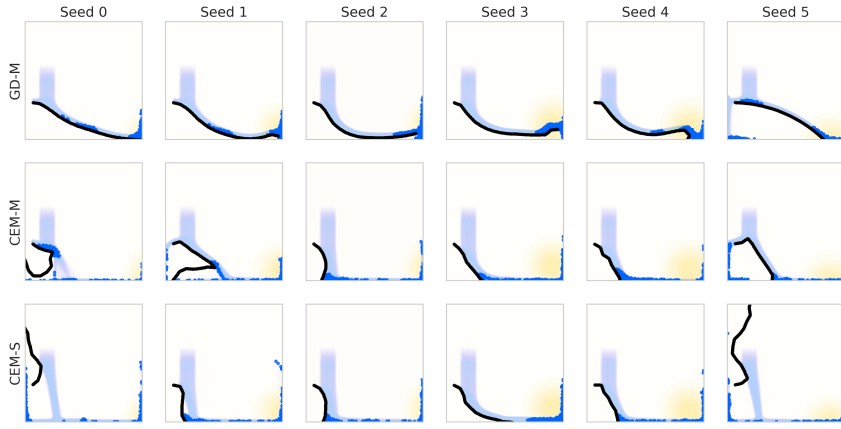

Figure A.15: Example solutions on *Ramp* task for each optimizer across 6 random seeds.

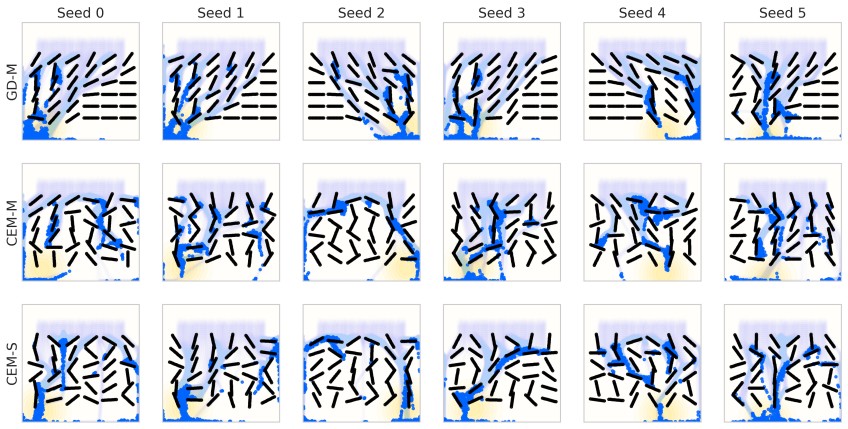

Figure A.16: Example solutions on *Maze* for each optimizer across 6 random seeds.

### D.5 End2End Surrogate Reward Models

#### D.5.1 3D WATERCOURSE

In this section we trained an end-to-end surrogate to map design parameters ($\phi_{\mathrm{map}}$) directly to rewards on the same designs as those contained in the training set for the GNN model (Section C.2), but extended with the rewards predicted by the ground truth simulator in each of the $\mathbf{d} = 8$ target angles of the *Direction* task, so that $\mathcal{D}_{sur} = \{(\phi_{\mathrm{map}}, f_{R_{\mathrm{dir}}}(\phi_{\mathrm{map}}, \mathbf{d}))^{(i)}\}_{i=0}^{1000}$. We used a ResNet-18 as the reward predictive model with a $25 \times 25$ dimensional input corresponding to the heightfield mesh, and we trained a separate model for each target direction.

The model could fit the training set, as well as an in-distribution test set very well (MSE=$0.0070 \pm 0.0025$). However, the model performs predictably poorly out of distribution (Figure 6): It (1) severely underestimates the reward of high-scoring designs found using GD-M (predicted=$0.33\pm0.06$,

actual=$1.21 \pm 0.18$, MSE=$0.813 \pm 0.344$), and (2) obtains poor designs when using this model in an GD optimizer (predicted=$1.12 \pm 0.31$, actual=$0.40 \pm 0.05$, MSE=$0.615 \pm 0.403$).

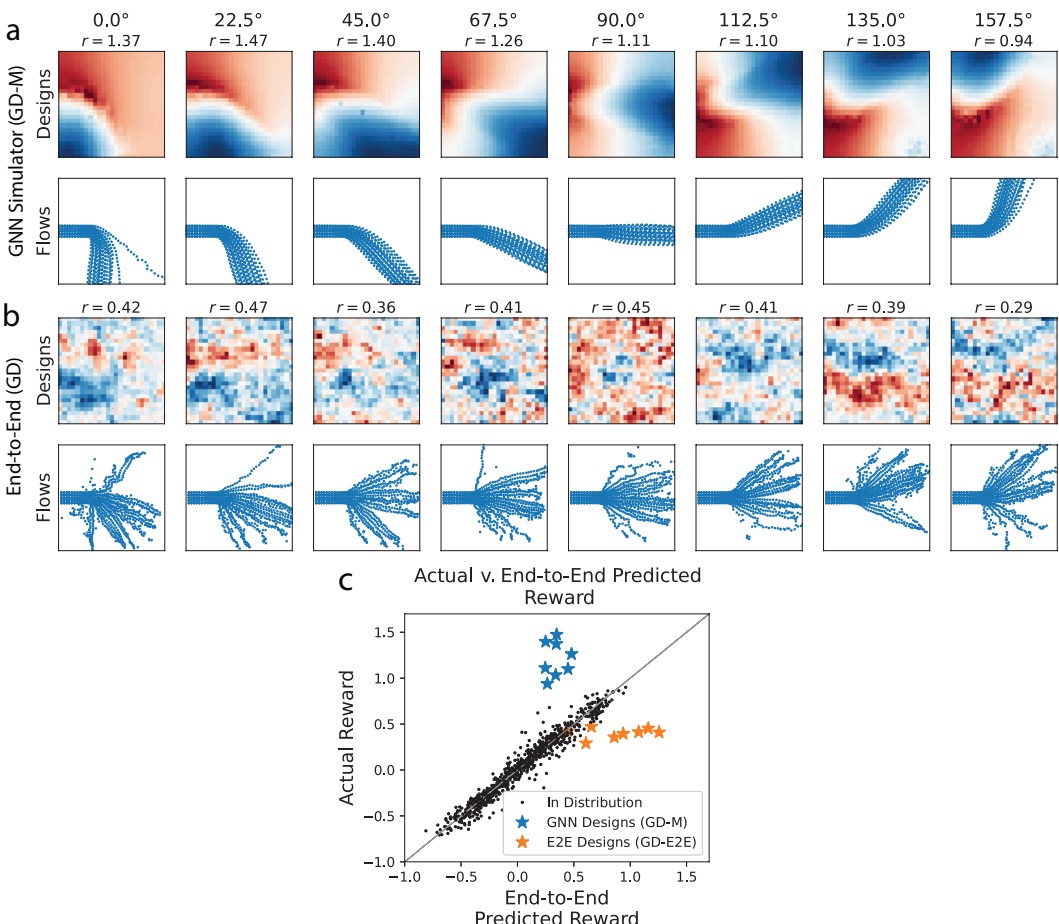

Figure A.17: Comparison of the end-to-end surrogate model (GD-E2E), which consists of a ResNet-18 [39] trained to predict reward directly from $25 \times 25$ grid of design parameters, on the 3D WATERCOURSE *Direction* task. (a - top row) an example of a design solution optimized with the Learned GNN Simulator (GD-M) for each direction. (a - bottom row) top view of the fluid flow for each solution using the design pictured above. (b - top row) an example of a design solution optimized using the trained End-to-End surrogate model (GD-E2E) for each direction. (b - bottom row) top view of the fluid flow for each solution using the design pictured above. (c) Ground Truth (evaluated using $f_s$) vs Predicted reward for different sets of designs: (black dots) held-out test data sampled from the same distribution used to sample training data, (blue stars) optimized designs shown in (a) found using the Learned GNN Simulator (GD-M), (orange stars) optimized designs shown in (b) found using the End-to-End model (GD-E2E).

### D.5.2 2D FLUID TOOLS

Unlike the dataset used in the previous section, the WATERRAMPS dataset does not share a design space with the 2D FLUID TOOLS tasks (i.e. there are no joints, only a small number of line segments), so we were unable to train a reward model end-to-end as above on this data. Therefore, we instead computed the reward for 1000 random designs (the same size as the training set for the GNN model) for the 2D FLUID TOOLS-*Contain* task using 4 up to 48 design parameters (Table A.2) to assess the difficulty of creating a dataset which includes high rewards using random sampling.

The maximum reward found decreased quickly for > 12 design parameters: i.e., for 48 joints, the highest reward encountered is $3\times$ less than the average reward design obtained using the GD-M

| Number of joints | 4 | 8 | 12 | 16 | 20 | 24 | 28 | 32 | 36 | 40 | 44 | 48 |
|---|---|---|---|---|---|---|---|---|---|---|---|---|
| Random design reward | 12.5 | 11.4 | 9.8 | 8.6 | 8.0 | 6.9 | 5.8 | 5.5 | 5.2 | 4.7 | 4.8 | 4.4 |

Table A.2: Maximum reward found via random exploration of 1k designs across different joint dimensions on the 2D FLUID TOOLS-*Contain* task.

(Figure 4-b) for the same dimensionality. Hence, a reward model trained on such data would not perform very well, as effective designs are highly out-of-distribution.

### D.6 Further tasks for 3D WATERCOURSE

To demonstrate the flexibility and further scaleability of the approach, we designed two further tasks for the 3D WATERCOURSE environment that require optimizing **two** meshes in order to manipulate the flow of a fluid: DOUBLEMESHREROUTE and DOUBLEMESHRAMP. Unlike in the original 3D WATERCOURSE tasks, in these two tasks the global rotation $\theta_g$ (in the $z$ plane) of the meshes can be optimized as well as the heightfields, leading to a design dimensionality of 1252 (each mesh has 626 parameters to optimize). Both tasks are particularly out of distribution for this environment, since there was only ever one mesh (with a sinewave heightfield) without any vertical rotation seen during training.

In both cases, the positions of the meshes are fixed to $(-0.25, -0.1, 0.0)$ and $(0.7, -0.2, 0.0)$ respectively, allowing them to be offset in both the height and distance from the water emitter.

#### D.6.1 DOUBLEMESHREROUTE task

In this task, the reward region is placed at $(0.5, 0.0, 0.5)$, which is just underneath the edge of the first mesh. In order to move the fluid to this location, the fluid needs to be guided by the first mesh into a stream, and then rerouted by the second mesh to bounce back and underneath the first mesh. We define the reward as the Gaussian probability for each particle $\mathbf{u_v}$ under $\hat{\mu}$ with $\mu = (0.5, 0.0, 0.5)$ (a region at $(0.5, 0.5)$ on the floor), i.e.

$$f_{R_{\mathrm{reroute}}} := \mathrm{mean}_v(\mathcal{N}(\mathbf{u_v}; \hat{\mu}, \sigma)) - \gamma_R \, \mathrm{mean}(\nabla \phi_{\mathrm{map}})$$

with $\sigma = 0.1$ and a regularization term $\gamma_R = 100$ to provide smoothness, as in the original 3D WATERCOURSE tasks.

The mesh rotations are initialized to $20°$ and $-40°$ respectively, and then allowed to be optimized further along with the heightfields (rotations end up at $35.6°$ and $-44.2°$ respectively after optimization). After 100 optimization steps, in this task the designs achieve a reward of 1.16 when evaluated with the ground truth simulator.

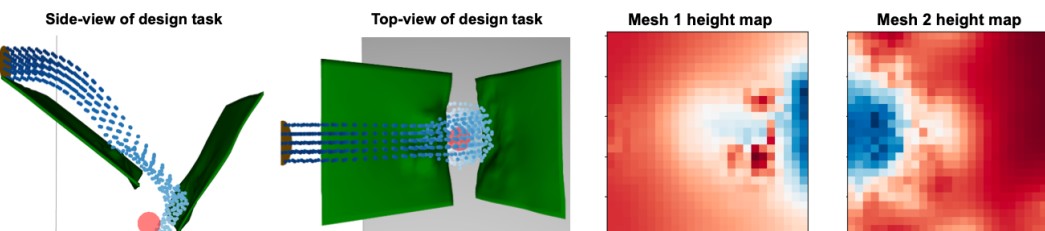

Figure A.18: Double mesh task where both the global rotation and the heightfield of **two meshes** can be optimized, a total of $625 \times 2 + 2 = 1252$ parameters. The model is the same one as used for the Direction and Cluster templates. This design task involves rerouting the fluid to the red ball region. To do so, the meshes must cooperate to reroute the fluid against its direction of travel using steep rotations. The mesh was never rotated in the training data, and only one mesh was ever seen in the training data.

### D.6.2 DOUBLEMESHRAMP task

In this task, the reward region is placed at $(2.5, 0.0, 0.5)$, which is past the edge of the second mesh. In order to move the fluid to this location, the fluid needs to be guided by the first mesh into a stream, and then rerouted by the second mesh to bounce back and underneath the first mesh. We define the reward as the Gaussian probability under $\hat{\mu}$ with $\mu = (2.5, 0.0, 0.5)$, i.e.

$$f_{R_{\text{ramp}}} := \text{mean}_v(\mathcal{N}(\mathbf{u}_\mathbf{v}; \hat{\mu}, \sigma)) - \gamma_R \, \text{mean}(\nabla \phi_{\text{map}})$$

with $\sigma = 0.1$ and a regularization term $\gamma_R = 100$.

The mesh rotations are initialized to $0°$, and then allowed to be optimized further along with the heightfields (rotations end up at $-4.4°$ and $-7.8°$ respectively after optimization). After 100 optimization steps, in this task the designs achieve a reward of $0.52$ when evaluated with the ground truth simulator.

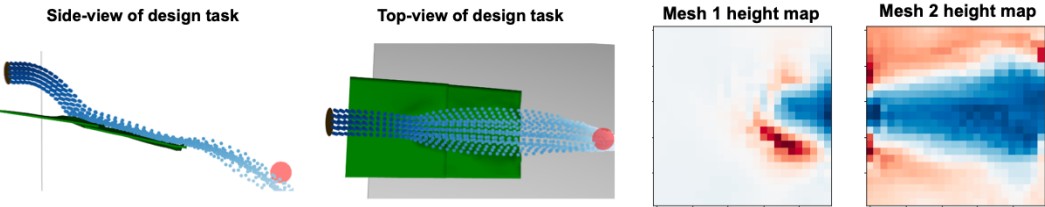

Figure A.19: Double mesh task where both the global rotation and the heightfield of **two meshes** can be optimized, a total of $625 \times 2 + 2 = 1252$ parameters. The model is the same one as used for the Direction and Cluster tasks. This design task involves rerouting the fluid to the red ball region. To do so, the meshes must cooperate to extend a ramp so that the water makes it all the way to the far side. Only a single, non-rotated mesh was seen in the training data.