# OpenReview forum: "Inverse Design for Fluid-Structure Interactions using Graph Network Simulators"
_NeurIPS.cc/2022/Conference — NeurIPS 2022 Accept_

### Official Review · Reviewer_p9AQ · 2022-06-30

**Rating:** 7
**Confidence:** 4
**Soundness:** 3 good
**Presentation:** 3 good
**Contribution:** 2 fair

**Summary:**

The authors investigate the performance of learned GNN simulators for inverse design.

**Questions:**

* GNNs simulators are chosen here without much explanation other than "they work".  Do you think the good performance of GNNs here might be due to tasks and dynamics involved, rather than some intrinsic (unexplained) superiority of the method? As the core focus of the  paper is performance in downstream design with learned simulators, an analysis of how other architectures used as surrogate solvers (for example, neural operators (1, 2, 3, 4, 5) ) compare would have been appreciated, particularly for the airfoil example.
* Why has CEM been chosen as the only baseline? The empirical performance of GNN simulators is the entire focus of this work, and any claim made would be stronger with at least another gradient-free optimization technique. I would appreciate comparisons with at least a single simple baseline from reinforcement learning, since it is mentioned in the introduction.

(1) Li et al., Fourier Neural Operator for Parametric Partial Differential Equations

(2) Guibas et al., Adaptive Fourier Neural Operators: Efficient Token Mixers for Transformers

(3) Gupta et al., Multiwavelet-based Operator Learning for Differential Equations

(4) Poli et al., Transform Once: Efficient Operator Learning in Frequency Domain

(5) Tran et al., Factorized Fourier Neural Operators

**Limitations:**

The authors discuss some limitations of the method. Some suggestions are provided in **questions** above (lack of baseline comparisons, no comparison with other types of learned simulators, and thus limited analysis on which tasks are appropriate for the approach and which require different models).

**Strengths And Weaknesses:**

**Strengths:**

* The approach is appealing in its simplicity: using a fixed, trained surrogate solvers for inverse problems, rather than relying on devising "in-the-loop" strategies to fine-tune the solver during solution of downstream tasks.
* The experiments are varied and interesting. The authors evaluate the model across particle-based fluid simulations as well as field-based steady-state flow around airfoils.

**Weaknesses:**

* From the perspective of advancing state-of-the-art (empirical or theoretical), the paper is not particularly satisfying due to a lack of contextualization of the proposed results. Part of it is certainly due to the fact that the tasks chosen involve a large amount of moving pieces. The authors provide ablations and helpful comments; however, the experimental section of the paper reads more like a post-hoc description of "we did this specific thing and it worked".
* I agree that showcasing GNN surrogate solvers on inverse problems is interesting. I also agree that this is a high-impact and open area of research. I do not think the experiments provided here do a good enough job at motivating the choice of models and baselines. The paper could benefit from adding targeted, designed "small scale" experiments aimed at clarifying why a specific choice made in the approach is appropriate.
* Only a single baseline (CEM) is considered.
* It would be of great value to the community if the code was open-sourced, particularly for an empirical work such as this one. Even with a detailed appendix, I do not believe the results are completely reproducible as is, especially given the amount hyperparameters involved (in both models and ground-truth simulators such as OpenFOAM). The authors attempt, wherever possible, to provide rules of thumb and empirical takeaways, which is appreciated.

---

> ### Author Response · Authors · 2022-08-02
> **Author response 1**
>
>
> Thank you for the thoughtful review, and for noting that this is an important area of work and that our experiments are varied and interesting. We respond to comments below and include a few clarification questions to try to improve the paper as much as possible.
>
> > “From the perspective of advancing state-of-the-art (empirical or theoretical), the paper is not particularly satisfying due to a lack of contextualization of the proposed results. Part of it is certainly due to the fact that the tasks chosen involve a large amount of moving pieces. The authors provide ablations and helpful comments; however, the experimental section of the paper reads more like a post-hoc description of "we did this specific thing and it worked".
>
> Would it be possible to clarify what kind of contextualization would help, and how the tasks involve a large number of moving pieces? We chose the experiments specifically to test various aspects of a learned simulator that would be critical to support design: generalization outside of the training data, stability over long rollouts, simulator runtime, scalability across different dimensions of task complexity, and the contribution of different modelling choices. We would very much appreciate more pointers for how to contextualize this better within the text to make our empirical contributions clearer (demonstrating that data-driven design can be accomplished with task-agnostic GNN-based simulators).
>
> > “I agree that showcasing GNN surrogate solvers on inverse problems is interesting. I also agree that this is a high-impact and open area of research. I do not think the experiments provided here do a good enough job at motivating the choice of models and baselines. The paper could benefit from adding targeted, designed "small scale" experiments aimed at clarifying why a specific choice made in the approach is appropriate.”
>
> Our targeted “small scale” experiments were the ablations we ran for the 2D Fluid Tools domain (Figure 4). In these ablations, we point out why specific choices for the GNN, such as the addition of noise during training, and the use of relative features, is critical for the performance of the approach. Are there other particular experiments you would like to see?
>
> > “It would be of great value to the community if the code was open-sourced, particularly for an empirical work such as this one. Even with a detailed appendix, I do not believe the results are completely reproducible as is, especially given the amount hyperparameters involved (in both models and ground-truth simulators such as OpenFOAM). The authors attempt, wherever possible, to provide rules of thumb and empirical takeaways, which is appreciated.”
>
> We are actively working on releasing the code for the paper. We note that much of the code is already open sourced: Our model is based on MeshGraphNets, for which code is available [here](https://github.com/deepmind/deepmind-research/tree/master/meshgraphnets). We use the 2D fluid tools training data from GNS [here](https://github.com/deepmind/deepmind-research/tree/master/learning_to_simulate), the Airfoil dataset is an adaptation of an [example](https://github.com/deepmind/deepmind-research/tree/master/learning_to_simulate) for DAFOAM, which is available [here](https://github.com/mdolab/dafoam), and the simulator for our 3D fluids tasks is available [here](https://github.com/InteractiveComputerGraphics/SPlisHSPlasH). Before the camera ready deadline, we will also release the simulation model implemented in JAX, and an end-to-end demonstration for using it for design optimization.

---

> > ### Author Response · Authors · 2022-08-02
> > **Author response 2**
> >
> > > “GNNs simulators are chosen here without much explanation other than "they work". Do you think the good performance of GNNs here might be due to tasks and dynamics involved, rather than some intrinsic (unexplained) superiority of the method? As the core focus of the paper is performance in downstream design with learned simulators, an analysis of how other architectures used as surrogate solvers (for example, neural operators (1, 2, 3, 4, 5) ) compare would have been appreciated, particularly for the airfoil example.”
> >
> > We did not intend to claim that GNNs are always the best choice for inverse design; FNOs could also be an excellent choice for solving inverse problems, particularly for continuous dynamics. We do however note a few interesting properties that are very useful for the type of dynamics that we study in this paper. Our experiments focus on fluid/structure interactions, with complex boundaries and/or free surfaces. This is a very important class of problems in design, as often boundary geometry is what is being designed, yet this is very much underexplored in ML for design. GNNs have a lot of representational flexibility and can model anything from particle-based sparse free-surface dynamics to continuous dynamics on meshes, making it easy to inject inductive biases. They have also been shown to provide plausible dynamics over both long rollouts and strong generalization over unseen geometries [55, 61], which is crucial for tasks involving designing geometry. Hence we believe they are a good fit for such fluid-structure design tasks.
> >
> > FNOs and spectral methods in particular have shown huge performance gains on PDEs with __limited__ boundary complexity. However, Fourier decomposition becomes more computationally expensive, and the features less informative, as boundary complexity increases (even though learned methods might be able to partially compensate for this). As such they might not be the best fit for the design tasks studied in this paper (e.g. particle-based or free-surface flows and dynamics driven by complex boundaries). However, we believe there is a place for both kinds of models in inverse methods generally. We decided to better clarify the scope and intention of our method in the text, and are changing the title to __Inverse Design for Fluid-Structure Interactions using Graph Network Simulators__.
> >
> > > “Only a single baseline (CEM) is considered / Why has CEM been chosen as the only baseline? The empirical performance of GNN simulators is the entire focus of this work, and any claim made would be stronger with at least another gradient-free optimization technique. I would appreciate comparisons with at least a single simple baseline from reinforcement learning, since it is mentioned in the introduction.”
> >
> > As noted on line 163, “We compare to further optimizer baselines, such as Bayesian Optimization [66] and CMA-ES [37] in the Appendix (Section D.2).” CEM was chosen as a commonly used method in model-based reinforcement learning, while Bayesian optimization and evolutionary CMA-ES are popular methods for design optimization. All approaches perform much worse than GD for the higher dimensional 3D Watercourse domain. As we outlined in the related work section, model-free RL or model-based RL methods which learn a policy alongside a model are inappropriate for the setting considered in this paper, as they require directly training on the task (or a proxy task), which is not available for any of our design problems.

---

> > > ### Comment · Reviewer_p9AQ · 2022-08-05
> > > **Response to authors**
> > >
> > > Thank you for the detailed response, which addresses most of my concerns. I support acceptance of this work and have raised my score to a 7. I also believe the new title reflects the contents of this paper much better.
> > >
> > > > Would it be possible to clarify what kind of contextualization would help, and how the tasks involve a large number of moving pieces?
> > >
> > > There are very few details on the reasoning behind the choice of base solvers (as far as I can see). As these are used in evaluating different methods to solve your inverse design tasks, it's unclear how relevant some of these results are in general, especially for runtime or speedup considerations. In particular, you choose solvers proposed in [43] and [8] for the 2D Fluid Tools and 3D Watercourse experiments without mentioning why those are meaningful to compare to. Not all solvers are created equal! As a concrete example, it would be great to check the performance of cheaper and less accurate solvers in combination with baseline methods for the inverse design task against surrogate solver + GD.
> > >
> > > Note that I am not crazily suggesting that you should have included every possible solver, or that this is a major limitation of the paper. I am merely suggesting that some more details on the properties of these solvers and why they make for representative baselines on your tasks would go a long way in making this paper more accessible and convincing to readers with different backgrounds. This in my opinion is even more important in papers that contain a large number of different examples.
> > >
> > > > As we outlined in the related work section, model-free RL or model-based RL methods which learn a policy alongside a model are inappropriate for the setting considered in this paper, as they require directly training on the task (or a proxy task), which is not available for any of our design problems.
> > >
> > > I do not agree. Your inverse design tasks can surely be cast as a reinforcement learning problem (not necessarily solved using model-based methods)? Environment: base solver evaluating "goodness" of current design, actions: design task actions. You would then be comparing in overall runtime and performance of surrogate solver + GD versus RL + base solver.

---

> > > > ### Author Response · Authors · 2022-08-07
> > > > **Thank you for your reply - a couple of follow-up questions.**
> > > >
> > > >  > Thank you for the detailed response, which addresses most of my concerns. I support acceptance of this work and have raised my score to a 7. I also believe the new title reflects the contents of this paper much better.
> > > >
> > > > Thank you! And thank you for the further very helpful comments below. We have a few follow-up questions that we would like your opinion on to further improve the paper.
> > > >
> > > > > There are very few details on the reasoning behind the choice of base solvers (as far as I can see). As these are used in evaluating different methods to solve your inverse design tasks, it's unclear how relevant some of these results are in general, especially for runtime or speedup considerations. In particular, you choose solvers proposed in [43] and [8] for the 2D Fluid Tools and 3D Watercourse experiments without mentioning why those are meaningful to compare to. Not all solvers are created equal! As a concrete example, it would be great to check the performance of cheaper and less accurate solvers in combination with baseline methods for the inverse design task against surrogate solver + GD.
> > > >
> > > > > Note that I am not crazily suggesting that you should have included every possible solver, or that this is a major limitation of the paper. I am merely suggesting that some more details on the properties of these solvers and why they make for representative baselines on your tasks would go a long way in making this paper more accessible and convincing to readers with different backgrounds. This in my opinion is even more important in papers that contain a large number of different examples.
> > > >
> > > > Thank you for this clarification. We will try to make this clearer in the text. We thought CEM was one of the cheaper alternatives since it does not require gradients, and we investigated how it changes as a function of the population size in the supplement (smaller populations -> cheaper evaluations). Do you have particular cheaper/less accurate solvers in mind that we could compare to?
> > > >
> > > > > I do not agree. Your inverse design tasks can surely be cast as a reinforcement learning problem (not necessarily solved using model-based methods)? Environment: base solver evaluating "goodness" of current design, actions: design task actions. You would then be comparing in overall runtime and performance of surrogate solver + GD versus RL + base solver.
> > > >
> > > > Right, we could do this, but one major advantage of GD + surrogate solver is that we never access the environment during optimization. Assuming base solver means the ground truth environment, RL+base solver would be trivially worse in terms of number of environment calls. In our case, the environment is also slower to run than the surrogate solver, which would mean that the runtime would also be worse. Our CEM-S comparisons which use the real environment for optimization show that using the ground truth model also does not seem to provide significant advantages in terms of overall performance. We can talk about some of these points more clearly in the experiments section, but does this explanation make sense, or have we still misunderstood your point?

---

### Official Review · Reviewer_oQtT · 2022-07-11

**Rating:** 5
**Confidence:** 4
**Soundness:** 4 excellent
**Presentation:** 4 excellent
**Contribution:** 2 fair

**Summary:**

In this work, the authors applied a learned physical simulator MESHGRAPHNETS [55] to physical design.

MESHGRAPHNETS is a graph network that can learn and predict the interaction among local particles (in fluids) or nodes (in meshes). Since the dynamics are described by a neural network, the simulation sequence is naturally differentiable when the time series is rolled out. Therefore, people can define some task-related objective functions on the trajectory and optimize the boundary condition using gradient-based methods.

Experiments demonstrate that this strategy works work even with the learned physical simulator. Although neural networks are prone to overfitting, and the long timestep could easily result in error accumulation, the experiments show that designs optimized on the MESHGRAPHNETS simulator work pretty well on ground-truth simulators.



**Questions:**

Generally, this paper writes well and I only have several minor questions.

1. In L279,

"However, since the learned simulator was trained on simpler data where these effects are unobserved, it picks up only on the appropriate collision performance and not the unrealistic edge cases. Thus, the learned simulator produces more plausible rollouts than the classical simulator in these cases, and might therefore be a better candidate for producing designs that would transfer to the real world."

It is common that learned physics can have low fidelity in uncovered corner cases. The corner cases are a major issue for other domains like autonomous cars. I wonder in the experiments whether the inaccuracy helps more or causes more problems for the designing process.

2. What is the definition of theta_R in L142? Is it defined somewhere before?

**Limitations:**

As I said in the Strengths And Weaknesses part, I think the technical contribution of this paper is limited. I suggest, if possible, doing some more challenging experiments or proposing some new algorithms.

1. Is it possible to involve some control policy networks within the open loop of design? Some new RL algorithms could be introduced in such a pipeline.

2. Can this method deal with larger-scale or more realistic scenarios? Like 3D Airfoil or more realistic fluids experiments (the particles in the fluids videos looks very toy demo).

3. Can this method be applied to the real world? A 3D printed watercourse or a 2D 'fluid' tool on a table with marbles can be very cool and much more convincing.

**Strengths And Weaknesses:**

I honestly like the message and ideology in this paper very much. However, the biggest concern for me is that the novelty of this paper itself is not that much.

Strengths

First, the paper is well-written and easy to follow. The motivation for conducting gradient-based design optimization using the learned physics simulator makes total sense. There have been a lot of previous methods that did this as well.

Second, although the experiment settings are simple, the results of the optimization results are impressive. It is known that prior learned surrogate physics models (I have also played with some of them) usually generalize poorly to out-of-domain data, let alone in a long simulation sequence where errors accumulation can be significant. But this paper demonstrates that MESHGRAPHNETS [55] performs pretty well and can optimize a design that is reasonably well in the ground-truth simulator.

Weaknesses

Nevertheless, in my view, the good results demonstrated in this paper should be credited to MESHGRAPHNETS. This paper simply wraps MESHGRAPHNETS into an optimizer package and then gets good results. As mentioned in L72, utilizing a differentiable learned module to do design optimization is very common. For the tasks in this submission, it is the design of MESHGRAPHNETS that makes this end-to-end optimization stable, generalizable, scalable, and efficient. The authors did a good verification for MESHGRAPHNETS but I hope to see more original and novel techniques.

For example, it would be good if the authors can show more challenging tasks, for example, training a control policy in the learned simulation which can instead be deployed in the ground-truth simulator or even the real world.

---

> ### Author Response · Authors · 2022-08-02
> **Author response 1**
>
> Thank you for the thoughtful review. Below we include some responses to weaknesses and questions which we hope to hear from you about.
>
> > “The motivation for conducting gradient-based design optimization using the learned physics simulator makes total sense. There have been a lot of previous methods that did this as well.”
>
> Could you point us to which papers you are referring to? Most of the papers we have found on design optimization do not learn the physics simulator, but instead learn a reward surrogate model that maps designs directly to rewards and differentiate through that model or assume a non-learned auto-differentiable model. L72 (mentioned below) explains that there are many machine learning approaches for gradient-based inverse design, but these do not optimize through a learned physics simulator to do so, and are therefore tied to a specific task and design space. What is important and conceptually novel about our work is that we do not need to learn a surrogate model that maps designs to rewards. Rather, we instead learn a model of the physical dynamics, which we use for optimization along with a design parameterization and reward function specified at test time. This is a fundamental distinction, as learning a dynamics model, rather than an end-to-end design->reward surrogate model, is what makes our approach task-agnostic.
>
> Differentiating through a learned dynamics model is sometimes applied in the context of model predictive control; however, this application is very different because of the option to update plans during the trajectory. Most methods must use much shorter rollouts (on the order of 3-10 steps, contrasted with our 10s to 100s)), and are typically investigated only for very small action spaces (on the order of 6-20 degrees of freedom, as opposed to our 10s-100s of design parameters). We are not aware of any other papers that differentiate through learned dynamics models in such complex state and action spaces for design problems, but would be happy to discuss references to papers we may have missed.
>
>
> > “Nevertheless, in my view, the good results demonstrated in this paper should be credited to MESHGRAPHNETS. This paper simply wraps MESHGRAPHNETS into an optimizer package and then gets good results. As mentioned in L72, utilizing a differentiable learned module to do design optimization is very common. For the tasks in this submission, it is the design of MESHGRAPHNETS that makes this end-to-end optimization stable, generalizable, scalable, and efficient. The authors did a good verification for MESHGRAPHNETS but I hope to see more original and novel techniques.”
>
>
> We emphasize that the paper’s goal is not to present a new model architecture. Instead, our contribution lies in demonstrating, to our knowledge for the first time, that design on complex domains can be achieved by differentiating through rollouts of a learned GNN simulator model. Importantly, this supports defining _completely new_ design spaces and reward functions at test time, which cannot be easily done with previous approaches to machine learning for design. We also outline the conditions that make this work -- such as relative encoding and training noise to stabilize gradients, which not every GNN has. We want to emphasize that these are by no means obvious findings; many black-box dynamics models fail when used in conjunction with an optimizer even when they demonstrate strong generalization properties in the forward direction, as optimizers can exploit even minuscule model delusions. We think that being able to perform stable inverse design by backpropagating through hundreds of model steps is actually quite a surprising result, certainly not shown by MeshGraphNets, and one that will open up new avenues for design using learned models. As other reviewers point out, we believe that this will be an impactful contribution to both the ML and computational design community.

---

> > ### Author Response · Authors · 2022-08-02
> > **Author response 2**
> >
> > > “For example, it would be good if the authors can show more challenging tasks, for example, training a control policy in the learned simulation which can instead be deployed in the ground-truth simulator or even the real world.”
> >
> > Our design tasks are already more complicated than most control tasks in terms of the dimensionality of the design space: a standard control policy may have 6 degrees of freedom (i.e of an arm) which is typically optimized over a time horizon of 5-10 timesteps at a time, thus resulting in a total of 30-60 control parameters to optimize. Our 3D Watercourse environment has 625 design dimensions to optimize, which is significantly more complex. We also want to clarify that for all evaluations in the paper, we deploy the optimized designs in the ground-truth solver. As a further challenging experiment, we have included new results where we have doubled the design dimensionality for the 3D Watercourse environment by considering a task that uses two 625 dimensional meshes (see updated supplemental material section D.6). Using exactly the same learned model as we used for the original design tasks, we can still optimize this 1252 dimensional design task to successfully reroute water to fall below the first mesh, or create a connected ramp to move water to a faraway location.
> >
> > > “It is common that learned physics can have low fidelity in uncovered corner cases. The corner cases are a major issue for other domains like autonomous cars. I wonder in the experiments whether the inaccuracy helps more or causes more problems for the designing process.”
> >
> > The point we were trying to make was less that GNNs can ignore “flawed” corner cases and more about generalization. Learned models are often known to generalize badly outside of the training distribution, which is generally seen as a drawback compared to using handwritten simulators. But here is a case where the reverse is true: the ground truth simulator code has a bug that causes erratic behavior, but only when a large number of rigid bodies are present in the scene-- i.e. it generalizes poorly beyond the scenarios it was built for. This was not exposed at all in the training set, which only used up to 4 rigid bodies, so the model couldn’t have possibly learned this “corner case”. The learned model instead was able to produce plausible dynamics, by learning local rules, and correctly extrapolating to a larger number of bodies. Such compositional extrapolation from local rules is generally a good prior for physical dynamics, which often can be described well by local interactions (e.g. PDEs), but this is much less true for autonomous car data. So in this setting, this behavior is desirable. Of course this doesn’t mean there aren’t other corner cases where the model generalizes badly, and it’s hard to quantify the effect without a “perfect” ground truth simulator. But we found it interesting, as it speaks to the ability of GNNs to learn local rules and generalize. We will try to clarify this in the text.
> >
> > > “What is the definition of theta_R in L142? Is it defined somewhere before?”
> >
> > theta_R refers to the parameters that represent a reward function (for example, the mean and standard deviation of a 2D gaussian or the number, location, and size of the pools for the 3D Watercourse environment). We will clarify this further in the problem formulation section.
> >
> > > “Can this method deal with larger-scale or more realistic scenarios? Like 3D Airfoil or more realistic fluids experiments (the particles in the fluids videos looks very toy demo).”
> >
> > The 3D Watercourse environment is very large-scale compared to standard design tasks, which typically consist of 10-100 design dimensions (see e.g. Design-Bench, Trabucco et al, 2022).  Watercourse has 625 design dimensions and thousands of particles (or even 1252 design dimensions in the new experiments we added in Appendix D.6) . While we could have used an ever higher particle resolution for this task, it would have had little impact on the actual design problem. We also note that the ‘toy’ look may simply be an artifact of the rendering, which is intended to highlight the particle representation -- rendering water as a dielectric material with a ray-tracer would result in a more realistic look (as e.g. [here](https://sites.google.com/corp/view/learning-to-simulate), which uses a comparable state space), but this is completely disconnected from the realism of the underlying physics simulation, which is identical in both cases.
> >
> > > “Is it possible to involve some control policy networks within the open loop of design? Some new RL algorithms could be introduced in such a pipeline.”
> >
> > Here we focused specifically on design problems via graph neural networks. However, given the promising results, we believe extending our work to study new RL strategies that leverage pre-trained graph network models of physical dynamics is an exciting avenue for future work, but outside the scope of this paper.

---

> > > ### Comment · Reviewer_oQtT · 2022-08-05
> > > **Thanks for the authors' reponse**
> > >
> > > Thanks for the authors' response. First of all, I agree with the direction and high-level ideas in this paper. And the presentation is good. The only thing I'm concerned about is the novelty.
> > >
> > > The authors also said in their rebuttal, "our contribution lies in demonstrating.". This is in my view the biggest issue. What I felt after reading this paper is "It demonstrates that MESHGRAPHNETS works really well! But what is the contribution of this paper? What are the challenges/novelties of applying MESHGRAPHNETS to solving optimization problem?" It is intuitive (which is a good thing) to use learned dynamics to design the morphology or/and controllers (e.g. [1, 2, 3]).
> > >
> > > In summary, I think this paper is sound and the demonstration is good, and the only drawback might be the novelty. I generally like the paper's idea and will not oppose its acceptance. As I mentioned in my review, if this paper can deal with more challenging cases or propose some new techniques (e.g. an RL algorithms especially for the learned dynamics), it could be much stronger.
> > >
> > > [1] Learning-In-The-Loop Optimization: End-To-End Control And Co-Design of Soft Robots Through Learned Deep Latent Representations. 2019. NeurIPS
> > >
> > > [2] Co-Learning of Task and Sensor Placement for Soft Robotics. 2021. IEEE Robotics and Automation Letters
> > >
> > > [3] Soft Robot Control With a Learned Differentiable Model. 2020. RoboSoft

---

> > > > ### Author Response · Authors · 2022-08-07
> > > > **Novelty with respect to cited papers, and emphasis on importance of analyses and applications**
> > > >
> > > > Thanks for your reply.
> > > >
> > > > We disagree that the paper would be better if it introduced a new model architecture or algorithm. As a community, we produce an enormous number of model variations but very few papers studying them thoroughly. This holds back progress. Some of the most impactful papers have been primarily analyses of existing models / architectures; for example, the Lottery Ticket Hypothesis (Best paper, ICLR 2019), and Deep Reinforcement Learning at the Edge of the Statistical Precipice (Best paper, NeurIPS 2021).
> > > >
> > > > As we have stated, the challenges/novelties of applying MESHGRAPHNETS to solve design problems is both conceptual and technical. It is very unintuitive that MESHGRAPHNETS could provide stable gradients over hundreds of time-steps, thousands of design dimensions, and thousands of particles to perform optimization. Most applications pass gradients through 10 steps at best, and even then often have issues with vanishing or exploding gradients. Results like these have not been shown previously. While the reviewer should come away from this thinking MESHGRAPHNETS works really well, that is indeed the surprising and novel finding – it works well enough that it can support a completely new kind of approach to machine learning for design.
> > > >
> > > > Furthermore, though the approach we present is intuitive, the cited papers [1] and [2] do __not__ use learned dynamics models to design the morphology or/and controllers. [1] uses an __analytical__ differentiable simulator, and learns a differentiable controller function to control it. [2] similarly represents and learns sensor placement differentiably, but uses __analytical__ differentiable simulators for optimization. The reason they do not use a learned simulator is because learned simulators historically have not provided stable enough gradients for optimization. Our paper actually presents a really exciting future direction for these, as it means it is now possible to learn the simulator that these papers assume has to be derived analytically. [3] does learn a simulator, but the paper only uses it to optimize for a single step of control (to do trajectory following). Again, this is because most learned simulators do not have stable enough gradients to optimize through many steps. Our paper is the first demonstration that this is possible, which could lead to a much wider adoption of these techniques in the community.
> > > >
> > > > > In summary, I think this paper is sound and the demonstration is good, and the only drawback might be the novelty. I generally like the paper's idea and will not oppose its acceptance. As I mentioned in my review, if this paper can deal with more challenging cases or propose some new techniques (e.g. an RL algorithms especially for the learned dynamics), it could be much stronger.
> > > >
> > > > Just to clarify, what more challenging cases would you like to see? In our paper we presented tasks with 625 design dimensions and thousands of particles in 3D, and in our rebuttal we added further tasks for manipulating 3D fluids with 1250 designs dimensions. This is orders of magnitude more complex than existing design tasks and control problems, as we previously noted.

---

> > > > ### Comment · Reviewer_oQtT · 2022-08-09
> > > > **Discussion about the novelties and more challenging cases**
> > > >
> > > > Thanks for your clarification.
> > > >
> > > > > We disagree that the paper would be better if it introduced a new model architecture or algorithm. As a community, we produce an enormous number of model variations but very few papers studying them thoroughly. This holds back progress
> > > >
> > > > Sure, it is totally fine with me that this paper wants to highlight its empirical verification. And I also agree that concrete evaluations and comparisons are needed and should be appreciated. In the last response, the authors point out that [1, 2] have analytical differentiable simulators, which is a big difference (also claimed in L68 "they are typically narrow in application scope"), If this is a major advantage for the proposed method, this argument might be better supported with a comparison (e.g. with Difftaichi, I think Difftaichi can also simulate the fluids experiments).
> > > >
> > > > > Just to clarify, what more challenging cases would you like to see?
> > > >
> > > > As I mentioned in the first round, a 3D airfoil could be more interesting, although the 2D one is a standard benchmark used in many scenarios. Also, the current examples mainly focus on a single kind of dynamics (fluids). Environments with various dynamics [1] could be a more thorough and challenging test for this method, for example, designing the shape for stone skipping (rigid body + fluids).
> > > >
> > > > [1] Learning Particle Dynamics for Manipulating Rigid Bodies, Deformable Objects, and Fluids. ICLR 2019
> > > >
> > > > In summary, I lean towards acceptance from the very beginning and just write some suggestions from my perspective of view. It is totally fine that the authors have their own emphasis.

---

### Official Review · Reviewer_J3rt · 2022-07-11

**Rating:** 6
**Confidence:** 3
**Soundness:** 3 good
**Presentation:** 4 excellent
**Contribution:** 2 fair

**Summary:**

This paper approaches the problem of modeling fluids using GNNs. The paper then shows how using GNNs as differentiable simulators enables the design of components that help solve fluid manipulation tasks.

**Questions:**

* Just to confirm, a different pretrained model is used per domain, but the same one is used across tasks?  E.g. all tasks for the 3D Watercourse use the same pretrained model but not the same one as for the airfoil.
* While the GNN is learnt independent of the reward function, it does seem like knowledge of the final application is included in the trained data set. E.g. for the waterworks model, the data includes a randomized obstacle plane that was sampled from random rotations and sine wave deformations. This seems like the deformations were purposely made to be smooth. What would happen if you added shot noise to the training data? Also how do you know when your model is sufficiently general? This seems vital to the motivation of the paper.


**Limitations:**

A limitation is that the paper focuses on fluid dynamics but references the area domain as a more general physical design problem. This should be highlighted.

**Strengths And Weaknesses:**

## Strengths:

* The paper is well-written and follows a clear structure.
* The experiments section is particularly well-written. The Figures are clear and really show the strength of the approach and also highlight a good choice of experiments. The additional supplementary materials on the webpage were helpful.
* The GNN does seem to do a good job of modeling the dynamics of the fluids for the three different domains.

## Weaknesses:

* The strength of this paper is as an applications paper. As a result the novelty is limited to the applications and not to the approach. It is not clear how this particular surrogate model is novel when compared to GNS and MeshGraphNets. It is not necessarily the case that just submitting an applications paper is a problem, it is just that the novelty of the approach must be taken into consideration.
* As an applications paper with valuable examples, it would be important to the community to make these examples available. The code does not seem to be provided with the paper and this would seem like a serious issue when considering whether the community would benefit from this paper.


## Minor Weaknesses:

* The title seems a little misleading as it is especially general. The abstract is also very general “a task-agnostic approach to inverse design…”. There is no mention of the fact that all experiments are within the domain of computational fluid dynamics. The reason the approach is so successful seems to be because the GNNs nicely match the simulators and the relationships between the fluid particles. It would probably make sense to make this clearer with a more specific title/abstract.
* Line 280-283: This seems to be an artifact of the simulator and not necessarily a strength/robustness of the approach. This part of the paper reads as if the fact that the ML model did some “smoothing” means that the ML model manages to avoid predicting the edge cases where the simulator fails and therefore is a more robust model choice. However, the reality is the ML model is actually failing when it does not predict these simulator edge cases as the ML model has only seen data from the “flawed” simulator and has only learnt information about the real-world through the “flawed” simulator. Perhaps this point should be removed.

---

> ### Author Response · Authors · 2022-08-02
> **Author response 1**
>
> Thank you for the thoughtful review. We have addressed your comments and questions below.
>
> > The strength of this paper is as an applications paper. As a result the novelty is limited to the applications and not to the approach. It is not clear how this particular surrogate model is novel when compared to GNS and MeshGraphNets. It is not necessarily the case that just submitting an applications paper is a problem, it is just that the novelty of the approach must be taken into consideration.
>
> We agree that our paper builds on existing GNN models and does not present a new network architecture. However, the approach _is_ novel in the context of machine learning for design. Unlike prior work, we show (to our knowledge for the first time) that design on complex domains can be achieved by differentiating through rollouts of a learned GNN simulator model rather than requiring a task-specific reward surrogate model. We also outline the conditions that make this work -- such as relative encoding and training noise to stabilize gradients, which not every GNN has. We want to emphasize that these are by no means obvious findings; many black-box dynamics models fail when used in conjunction with an optimizer even when they demonstrate strong generalization properties in the forward direction, as optimizers can exploit even miniscule model delusions. We think that being able to perform stable inverse design by backpropagating through 100s of model steps is actually quite a surprising result, and one that will open up new avenues for design using learned models. As the reviewer points out, we believe that this will be an impactful contribution to both the ML and computational design community.
>
>
> > “As an applications paper with valuable examples, it would be important to the community to make these examples available. The code does not seem to be provided with the paper and this would seem like a serious issue when considering whether the community would benefit from this paper.”
>
> We are actively working on releasing the code for the paper. We note that much of the code is already open sourced: Our model is based on MeshGraphNets, for which code is available [here](https://github.com/deepmind/deepmind-research/tree/master/meshgraphnets). We use the 2D fluid tools training data from GNS [here](https://github.com/deepmind/deepmind-research/tree/master/learning_to_simulate), the Airfoil dataset is an adaptation of an [example](https://github.com/deepmind/deepmind-research/tree/master/learning_to_simulate) for DAFOAM, which is available [here](https://github.com/mdolab/dafoam), and the simulator for our 3D fluids tasks is available [here](https://github.com/InteractiveComputerGraphics/SPlisHSPlasH). Before the camera ready deadline, we will also release the simulation model implemented in JAX, and an end-to-end demonstration on how to use it for design optimization.
>
> > “The title seems a little misleading as it is especially general. The abstract is also very general “a task-agnostic approach to inverse design…”. There is no mention of the fact that all experiments are within the domain of computational fluid dynamics. The reason the approach is so successful seems to be because the GNNs nicely match the simulators and the relationships between the fluid particles. It would probably make sense to make this clearer with a more specific title/abstract.”
>
> We agree, and are instead suggesting the title “Inverse Design for Fluid-Structure Interactions using Graph Network Simulators” to make this clearer. We will also update the abstract to clarify that we focus on domains that involve fluids interacting with other structures.

---

> > ### Author Response · Authors · 2022-08-02
> > **Author response 2**
> >
> > > “Line 280-283: This seems to be an artifact of the simulator and not necessarily a strength/robustness of the approach. This part of the paper reads as if the fact that the ML model did some “smoothing” means that the ML model manages to avoid predicting the edge cases where the simulator fails and therefore is a more robust model choice. However, the reality is the ML model is actually failing when it does not predict these simulator edge cases as the ML model has only seen data from the “flawed” simulator and has only learnt information about the real-world through the “flawed” simulator. Perhaps this point should be removed.”
> >
> > Yes, this is indeed a subtle point. In some settings this could indicate a weakness of the learned model, by not learning to truly mimic the behavior of the simulator. We believe however this specific case illustrates an interesting point: Learned models are often known to generalize badly outside of the training distribution, which is generally seen as a drawback compared to using handwritten simulators. But here is a case where the reverse is true: the ground truth simulator has a bug that causes implausible behavior, but only when a large number of rigid bodies are present in the scene-- i.e. it generalizes poorly beyond the scenarios it was built for. This behavior was not exposed at all in the training set, which only used up to 4 rigid bodies, so the model couldn’t have learned the “flawed” behavior. The learned model instead was able to generalize to larger numbers of bodies and produce plausible dynamics, not by “smoothing” but by learning the correct, local physical rules.
> >
> > While this of course doesn’t mean there aren’t other corner cases where the model generalizes badly, and it’s hard to quantify the effect in the absence of a “perfect” ground truth simulator, we did find it interesting, as it does speak to the GNNs power to learn local rules to support generalization. We will try to clarify this in the text, but if the reviewers still feel it’s a distraction, we can remove it.
> >
> > > “Just to confirm, a different pretrained model is used per domain, but the same one is used across tasks? E.g. all tasks for the 3D Watercourse use the same pretrained model but not the same one as for the airfoil.”
> >
> > Yes, that is correct. A different pretrained model is used for each domain, but the model is fixed across all tasks within that domain. Furthermore, the architecture of the GNN is fixed across domains. No domain-specific hyperparameter tuning was needed.
> >
> > > “While the GNN is learnt independent of the reward function, it does seem like knowledge of the final application is included in the trained data set. E.g. for the waterworks model, the data includes a randomized obstacle plane that was sampled from random rotations and sine wave deformations. This seems like the deformations were purposely made to be smooth. What would happen if you added shot noise to the training data? ”
> >
> > Based on [55], [61], we expect that we would get good generalization even for very different obstacle geometries, provided that the model saw sufficient diversity at the local level in the training data. We observed this in the 2D fluids domain; the (external) training set contains interactions between the fluid and simple ramps, but with a very different distribution to our task (e.g. no curved shapes). We will likely need to see some smooth surfaces in the training set for WaterCourse, such that the model can learn local rules of fluid/surface interactions. Training purely on shot noise might not be the best choice for these and most other design tasks, as it would generate spiky meshes with a very uncharacteristic local geometry and unstable dynamics.
> >
> > However, we do not believe that training on perfectly smooth sine waves in particular is essential; e.g. a plane with Perlin noise, or on a set of geometries unrelated to the task (e.g. sphere, cube, teapot) should work as well. Unfortunately the short rebuttal window doesn’t give us enough time to design a dataset, re-train models and re-run optimization. For the camera ready, we plan to train a new dynamics model.
> >
> > To illustrate a higher degree of generalization to design tasks that are outside of the training distribution, we performed a new design experiment in the 3D Watercourse domain (see updated supplemental materials Appendix Section D.6). In this design setup, instead of only optimizing one mesh, we optimize two meshes, and control not just the surface height but also the global rotation of each mesh independently. The training data never saw meshes at any angle other than 0 degrees, nor did it ever see more than one mesh. However, in this design task with multiple, rotated meshes, we are still able to successfully optimize for a design that directs water into a pool centered below the first mesh (requiring fluid rerouting), as well as a design that can extend and shape a ramp to get fluid into a far away region by connecting two meshes.

---

> > > ### Author Response · Authors · 2022-08-02
> > > **Author response 3**
> > >
> > >
> > > > “Also how do you know when your model is sufficiently general? This seems vital to the motivation of the paper.”
> > >
> > > Estimating OOD generalization limits is very much an unsolved problem in ML; but empirically [55], [61] have shown that GNN physics models generalize well on composition, that is, if we ensure that the dataset exposes local interactions that allow the model to learn local physical rules, we have a good chance of it generalizing to a (potentially more complex) OOD scene with those dynamics. Of course, it is not a priori clear that effective forward predictions imply that a model is sufficiently general to support design optimization, which is why we tested this for graph network simulators across a selection of challenging design problems and analyzed different model choices that are important to make this happen (Figure 4-d).
> > >
> > > In terms of model training, we use standard techniques commonly applied in supervised learning; we use validation datasets with unseen initial conditions (sampled in the same way as the training data) and choose the model with the lowest rollout error.
> > > While this paper proposes a task-agnostic approach to design, we can also expect to improve model accuracy by training or fine-tuning on task-specific data if we assume this data is available.

---

> > > ### Comment · Reviewer_J3rt · 2022-08-05
> > > **Reviewer Response 2**
> > >
> > > >  In some settings this could indicate a weakness of the learned model, by not learning to truly mimic the behavior of the simulator.
> > >
> > > Thanks for the response on this point. It is a difficult one because it is definitely interesting that the model learnt a more "generalized" behaviour, but does that mean the model fails in mimicking the simulator (which it was designed to do). In other engineering applications, a corner case may actually be catastrophic and be very rarely sampled from a simulator and possibly glossed over by the model. In that scenario one would want the simulator to model the corner cases. I guess it ultimately depends on the application and the safety/risks involved. I don't think the authors need to remove this. Perhaps expanding on this point a bit more would be interesting if there is space.
> > >
> > > One other thought is that perhaps if you were to learn from multiple different simulators, then generalization would perhaps be something that one could claim for the model.
> > >
> > > > For the camera ready, we plan to train a new dynamics model.
> > >
> > > That would definitely be interesting!

---

> > > > ### Author Response · Authors · 2022-08-07
> > > > **These are great points - we will expand on them for the camera-ready.**
> > > >
> > > > These are great points, and we will make sure to include an expanded discussion of this with the extra page we are given for the camera-ready version.

---

> > ### Comment · Reviewer_J3rt · 2022-08-05
> > **Reviewer Response 1**
> >
> > Thanks for your detailed response!
> >
> > > We are actively working on releasing the code for the paper. We note that much of the code is already open sourced: Our model is based on MeshGraphNets, for which code is available here. We use the 2D fluid tools training data from GNS here, the Airfoil dataset is an adaptation of an example for DAFOAM, which is available here, and the simulator for our 3D fluids tasks is available here. Before the camera ready deadline, we will also release the simulation model implemented in JAX, and an end-to-end demonstration on how to use it for design optimization.
> >
> > I appreciate that some of the code is available, but it is still a big weakness that the code is not available yet. This is really the main weakness preventing me from providing a higher score, as the paper is otherwise of high quality. It is definitely important to make sure code release is a high priority for this work.

---

> > > ### Author Response · Authors · 2022-08-07
> > > **Code will be released**
> > >
> > > We agree that releasing the code is a top priority, hence, we have committed to doing so in this rebuttal period (which will be __made public__ to the NeurIPS community if the paper is accepted, and therefore holds us accountable to this promise). Concretely, we will  release the model implemented in JAX and an end-to-end demonstration on how to use it for design optimization.
> > >
> > > Given the short timeframe of the rebuttal phase, we placed more focus on running the experiments that reviewers asked for (including the ones investigating design tasks further from the training data for the 3D domain). Therefore, it was not feasible to clean up the code sufficiently for it to be released on such a short timeframe in a way that would be useful to the community. We hope this eases your concerns since you mentioned code availability as the only thing preventing a higher score.

---

### Official Review · Reviewer_zFDa · 2022-07-12

**Rating:** 7
**Confidence:** 3
**Soundness:** 3 good
**Presentation:** 4 excellent
**Contribution:** 3 good

**Summary:**

This paper proposes to 'invert' graph neural networks (GNNs) in the search for optimal solutions in the context of engineering design. The paper uses stochastic gradient descent for optimization and observes that the gradients can be reliably propagated in a coarse trajectory. The paper evaluates the method in different CFD tasks and shows significant advantages compared to sampling-based techniques used in model-based control. An important advantage of the method is its generalization to unseen inverse-design scenarios.

**Questions:**


- Line 108: The abbreviation GNS is not defined.
- Generalization in the 2D Fluid Tools is well explained (Lines 265-269). I could not find the same explanation for the airfoil setup. It is unlikely, but it appears that (Lines 198-205) the GNN is learned in a typical supervised way. I checked the supplementary materials and was left a bit confused. For example, speaking of a rollout of length 1 (supplementary materials Line 766) means the whole learning has been carried out without transitions specific to GNNs?

**Limitations:**

- I think the generality argument could have been better supported using a more systematic approach.
- The suitability of the forward model toward CFD tasks could be highlighted.

**Strengths And Weaknesses:**

STRENGTHS:
- The important observation of suitability of using GNNs for inverse design.
- The solid execution of the idea and an extensive set of applications in different scenarios.
- The writing.


WEAKNESSES:
- The lack of evaluation against a fully data-driven approach to inverse design. These models, which are learned from pairs of data in a supervised fashion (for example, MO-PaDGAN: Reparameterizing Engineering Designs for Augmented Multi-Objective Optimization), are arguably as important as the sampling methods used for comparison in the paper.
- The method is incremental in a straightforward way in the sense it observes the general behavior of GNNs and proposes gradient-based optimization in the context of inverse design. The strengths of the method (long trajectories, generalization) are native to the forward surrogate models (GNNs) introduced in earlier works and not because of special treatment of the inverse problem per se. This makes the paper more an application of the GNN simulations. However, given the broad implication, this should not prevent the paper from a presentation to a broad audience.
- The tile is unnecessarily broad. "Differentiable learned simulator" could pertain to a lot of different ML models (MLP, CNN, GP, etc.). Also, the applications seem well suited to computational fluid dynamics (CFD) tasks as shown in all 3 examples. In a more focused approach, this paper could have been called something along the lines: Inverse design in CFD using graph neural networks.

---

> ### Author Response · Authors · 2022-08-02
> **Author response 1**
>
> Thank you for the thorough and thoughtful review. We are very glad that you agree that the observations in the paper are both important and relevant to a broad audience. We provide responses to the weaknesses below, as well as answers to the questions.
>
> > “The lack of evaluation against a fully data-driven approach to inverse design. These models, which are learned from pairs of data in a supervised fashion (for example, MO-PaDGAN: Reparameterizing Engineering Designs for Augmented Multi-Objective Optimization), are arguably as important as the sampling methods used for comparison in the paper.”
>
> We agree that a comparison to a fully data-driven approach to inverse design could be a valuable addition to the paper. We have performed an additional experiment for this (described below) and added the results to the supplemental material (Appendix D.5), which can be worked into the main manuscript if you think it should be in the main text.
> However, we want to clarify that one of the main contributions of this paper is a conceptual shift towards _task-agnostic_ methods for design, by learning _simulator_ models at training time rather than _task-specific_ models mapping design to reward. We see several fundamental advantages to this approach:
> It allows us to change the task specification (i.e. reward) and design space at test time, without retraining the model; allowing for fast design iteration (for example, we tested two new design tasks for the 3D environment during the rebuttal phase without any further training, see responses to other reviewers below.)
> Simulator models have a much better chance of OOD generalization, since the physical dynamics they learn are often local and composable [55, 61]
> Fully data-driven methods require high-reward samples (i.e. something close to the desired design solution) to be present in the training set. This is a difficult exploration problem. For non-trivial design problems, creating a dataset with random sampling is not sufficient (see below). Instead, clever heuristics are often required for dataset creation, such as those presented in [MO-PaDGAN-AppendixD].
>
> In an __additional experiment (D.5.1)__, we trained a ResNet18 reward model on the training set for the 3D Watercourse-(Direction) task. The model directly maps the 25x25 design space to reward. The model was trained on the same dataset that our GNN dynamics model was trained on (random sinusoidal designs), annotated with the design rewards computed using the ground truth simulator. The model could fit the training set, as well as an in-distribution test set, very well (reward MSE=0.0070+/-0.0025). However, the model performs poorly out of distribution (Figure A.17). It (a) severely underestimates the reward of high-scoring designs found using our GNN simulator (predicted=0.33+/-0.06, actual=1.21+/-0.18, MSE=0.813+/-0.344), and (b) we obtain poor designs when using this model with a GD optimizer (predicted=1.12+/-0.31, actual=0.40+/-0.05, MSE=0.615+/-0.403), as the model suffers significantly more from extreme model delusions during optimization (see Figure A17.b, orange stars). While in principle more sophisticated methods could be used beyond GD, the fact that the reward model so poorly predicts high-performing designs from the GNN suggests that most approaches based on learning surrogate reward models would fare badly here.
>
> We also note that our task-agnostic approach provides more flexibility; for example, the GNN model for 2D FluidTools was trained on a pre-existing dynamics dataset. This dataset doesn’t share a design space with our tasks (i.e. there are no joints, only a small number of line segments), so we couldn’t even train a reward model end-to-end as above on this data.
>
> Finally, we performed an __additional experiment__ to assess the difficulty of creating a dataset which includes high rewards using random sampling (Appendix D.5.2). We computed the reward for 1000 random designs (the same size as the training set for the GNN model) for the 2D FluidTools-Contain task for 4-48 design parameters. Unsurprisingly, the maximum reward found decreased quickly for > 12 design parameters. For 48 parameters, the highest reward encountered is 3x less than the  design obtained using the GNN model. Hence, a reward model trained on such data wouldn’t perform very well, as effective designs are highly out-of-distribution.

---

> > ### Author Response · Authors · 2022-08-02
> > **Author response 2**
> >
> > > “The method is incremental in a straightforward way in the sense it observes the general behavior of GNNs and proposes gradient-based optimization in the context of inverse design. The strengths of the method (long trajectories, generalization) are native to the forward surrogate models (GNNs) introduced in earlier works and not because of special treatment of the inverse problem per se. This makes the paper more an application of the GNN simulations. However, given the broad implication, this should not prevent the paper from a presentation to a broad audience.”
> >
> > We agree that our paper builds on existing GNN models and does not present a new network architecture. Instead, our contribution lies in demonstrating, to our knowledge for the first time, that design on complex domains can be achieved by differentiating through rollouts of a learned GNN simulator model without the need to pre-train a reward surrogate. We also outline the conditions that make this work -- such as relative encoding and training noise to stabilize gradients, which not every GNN has. We want to emphasize that these are by no means obvious findings; many black-box dynamics models fail when used in conjunction with an optimizer even when they demonstrate strong generalization properties in the forward direction, as optimizers can exploit even miniscule model delusions. We think that being able to perform stable inverse design by backpropagating through 100s of model steps is actually quite a surprising result, and one that will open up new avenues for design using learned models. As you point out, we believe that this will be an impactful contribution to both the ML and computation design community.
> >
> > > “The tile is unnecessarily broad. "Differentiable learned simulator" could pertain to a lot of different ML models (MLP, CNN, GP, etc.). Also, the applications seem well suited to computational fluid dynamics (CFD) tasks as shown in all 3 examples. In a more focused approach, this paper could have been called something along the lines: Inverse design in CFD using graph neural networks.”
> >
> > Thank you for this suggestion. In hindsight we agree that a more specific title would make the contributions of this paper clearer, and are suggesting “Inverse Design for Fluid-Structure interactions using Graph Network Simulators” as an alternative. Does this seem suitable?
> >
> > > “Line 108: The abbreviation GNS is not defined.”
> >
> > Thank you for catching this. We will update the paper accordingly.
> >
> > > “Generalization in the 2D Fluid Tools is well explained (Lines 265-269). I could not find the same explanation for the airfoil setup. It is unlikely, but it appears that (Lines 198-205) the GNN is learned in a typical supervised way. I checked the supplementary materials and was left a bit confused. For example, speaking of a rollout of length 1 (supplementary materials Line 766) means the whole learning has been carried out without transitions specific to GNNs?”
> >
> > We apologize for the confusion. The GNN for airfoil is _not_ learned in a typical supervised way (i.e. directly mapping design parameters to reward). We instead learn a surrogate aerodynamics model; i.e. we learn to predict the RANS pressure and Reynolds stress fields discretized on a 4158 node simulation mesh. The input to the network is the wing geometry and inflow air velocity, discretized on the same mesh. This task-agnostic setup allows us, in theory, at test time, to use arbitrary design functions and reward functions over the pressure field without retraining the model.
> >
> > In contrast to the other dynamical simulation domains, this is a steady-state prediction problem, i.e. rollout length is always 1 by definition. However, we still perform several rounds of message passing within this one step. The benefits of the GNNs for physics modeling are not specific to dynamical simulation, and this task should also strongly benefit from inductive biases like translational and permutation equivariance. We will clarify this point in the text.

---

> > > ### Comment · Reviewer_zFDa · 2022-08-04
> > > **Rebuttal addresses my questions**
> > >
> > > Thank you for the thorough rebuttal. It addresses my questions. I am raising my score.
> > > Concerning comparison with an end-to-end strategy, I would include it if the space allows. I think the high generality is the most important message of this paper and highlighting it next to an end-to-end method could help with the impact of the paper. For a huge community outside ML, data-driven inverse design means fully, supervised end-to-end modeling. If they see an advantageous method next to what they know, the chances for adoption increase.

---

> > > > ### Author Response · Authors · 2022-08-05
> > > > **Thank you for raising your score!**
> > > >
> > > > With the extra page we have available for the camera-ready, we will move the analysis from the supplement to the main text based on your recommendation. Thank you again for your thoughtful and helpful review.

---

### Author Response · Authors · 2022-08-02
**Summary of changes and overall response**

We thank all reviewers for their thoughtful comments, and are glad that they found our work to be “solidly executed”, of “broad impact”, “well-written”, and that they agree with our “message and ideology”. We respond to the individual comments below; we hope to have addressed the concerns in these reviews. For the benefit of all reviewers and the AC, we summarize here major aspects of our response as well as additional experiments.

All reviewers rated the contribution of the paper as being “fair” with both “good” or “excellent” soundness and presentation. To reiterate our contributions: our paper builds on existing GNN models and does not present a new network architecture. Instead, our contribution lies in demonstrating, to our knowledge for the first time, that design on complex domains can be achieved by differentiating through rollouts of a learned GNN simulator model. Importantly, this supports defining __completely new__ design spaces and reward functions at test time, which cannot be done with previous end-to-end approaches to machine learning for design. We also outline the conditions that make this work -- such as relative encoding and training noise to stabilize gradients, which not every GNN has. We want to emphasize that these are by no means obvious findings; many black-box dynamics models fail when used in conjunction with an optimizer even when they demonstrate strong generalization properties in the forward direction, as optimizers can exploit even minuscule model delusions. We think that being able to perform stable inverse design by backpropagating through hundreds of model steps is actually quite a surprising result, and one that will open up new avenues for design using learned dynamics models. As at least one reviewer points out, we believe that this will be an impactful contribution to both the ML and computational design community.


To further support these contributions, and to better scope the paper, we have made these specific changes and ran these additional experiments:
1. To better clarify our specific contributions, we have changed the title of the paper to: __Inverse Design for Fluid-Structure Interactions using Graph Network Simulators__
2. We have demonstrated that a model trained “end-to-end” (that is, it learns a mapping directly from design parameters to reward) performs significantly worse than learning a model of the simulator dynamics for the 3D WaterCourse domain, struggling in particular because of its inability to generalize out-of-distribution as well as the learned simulator (__Appendix D.5.1__).
3. We have run additional baselines quantifying the maximum reward obtained by randomly sampling designs in the 2D task, showing that it is extremely unlikely to yield any designs with high reward. This highlights the challenging exploration problem in these tasks (__Appendix D.5.2__).
4. We have run design optimization for two additional, even more complex design experiments in 3D WaterCourse which are further from the training set (with multiple meshes, and including global rotation optimization; __Appendix D.6__). This underscores the generality of our approach – we simply defined a new design space and new reward function for optimization, without requiring costly model re-training.

We have uploaded a revised version of the paper with the above additions to the appendix. We will include this and all other more extensive text changes in the final version.

---

### Meta-Review · Area_Chair_G2DM · 2022-08-23

**Recommendation:** Accept
**Confidence:** Certain

**Metareview:**

This work proposes to use GNNs in acting as good simulators of fluid dynamics. It is a solid example of using neural networks as differentiable simulators that can then be used for solving for design of components for fluid manipulation tasks. Overall, this work has been well received and should inspire related work in inverse design. The authors are encouraged to take the detailed reviews into account for camera-ready and especially make sure to make high quality reproducible code available to the community. There has also been discussion about the naming of the paper to reflect proper scope and that should be considered as well.

**Award:**

No

---

### Decision · Program_Chairs · 2022-09-14

Accept